# Concerted oxygen diffusion across heterogeneous oxide interfaces for intensified propane dehydrogenation

Sai Chen[1,2,3], Ran Luo[1,2,3], Zhi-Jian Zhao [1,2], Chunlei Pei[1,2], Yiyi Xu[1,2], Zhenpu Lu[1,2], Chengjie Zhao[1,2], Hongbo Song[1,2] & Jinlong Gong [1,2,3,4] ✉

Propane dehydrogenation (PDH) is an industrial technology for direct propylene production which has received extensive attention in recent years. Nevertheless, existing non-oxidative dehydrogenation technologies still suffer from the thermodynamic equilibrium limitations and severe coking. Here, we develop the intensified propane dehydrogenation to propylene by the chemical looping engineering on nanoscale core-shell redox catalysts. The core-shell redox catalyst combines dehydrogenation catalyst and solid oxygen carrier at one particle, preferably compose of two to three atomic layer-type vanadia coating ceria nanodomains. The highest 93.5% propylene selectivity is obtained, sustaining 43.6% propylene yield under 300 long-term dehydrogenation-oxidation cycles, which outperforms an analog of industrially relevant K-CrO$_x$/Al$_2$O$_3$ catalysts and exhibits 45% energy savings in the scale-up of chemical looping scheme. Combining in situ spectroscopies, kinetics, and theoretical calculation, an intrinsically dynamic lattice oxygen "donator-acceptor" process is proposed that O$^{2-}$ generated from the ceria oxygen carrier is boosted to diffuse and transfer to vanadia dehydrogenation sites via a concerted hopping pathway at the interface, stabilizing surface vanadia with moderate oxygen coverage at pseudo steady state for selective dehydrogenation without significant overoxidation or cracking.

Propane dehydrogenation (PDH) is an industrially important alternative to oil-based cracking processes[1,2]. However, the commercial non-oxidative propane dehydrogenation containing CrO$_x$ or Pt-based catalysts is endothermic and equilibrium-limited, necessitating much heat to achieve viable propylene yield[3,4]. Although the oxidative dehydrogenation of propane (ODH) has the potential to improve conversion for favorable thermodynamics, propylene selectivity is hampered by overoxidation to CO$_2$[5,6]. A similar challenge is faced in selective oxidation reactions in the chemical industry[7,8].

Chemical looping engineering offers exciting new opportunities for the challenges through the physical or temporal separation of dehydrogenation and oxidation by solid oxygen carrier mediums[9,10]. Unlike traditional catalysts, the oxygen carriers react with alkanes and undergo reversible changes by donating and replenishing oxygen to close the loop in the reducer and oxidizer reactors. Most oxygen carriers involve the metal centers or oxide composites to modulate lattice oxygen reactivity, using bulk doping[11], surface modification[12], or confinement in supports[13]. Recently, vanadia/ceria catalysts have attracted increased attention in the oxidative dehydrogenation of propane with

[1]Key Laboratory for Green Chemical Technology of Ministry of Education, School of Chemical Engineering & Technology, Tianjin University, Tianjin 300072, China. [2]Collaborative Innovation Center for Chemical Science & Engineering (Tianjin), Tianjin 300072, China. [3]Joint School of National University of Singapore and Tianjin University, International Campus of Tianjin University, Binhai New City, Fuzhou 350207, China. [4]National Industry-Education Platform of Energy Storage, Tianjin 300350, China. ✉e-mail: jlgong@tju.edu.cn

$O_2$ co-feeding. The electronic effects and redox properties were investigated at the molecular level[13–16]. Nevertheless, direct experimental and theoretical insights into the lattice oxygen diffusion and the surface dynamics have not been reported yet for the anaerobic oxidative dehydrogenation via chemical looping engineering.

In this work, to unravel the oxygen diffusion and reaction dynamics regarding the active sites, a nanoscale core-shell redox catalyst combining dehydrogenation catalyst and oxygen carrier at one particle is designed. The core-shell redox catalyst is preferably composed of two to three atomic layer-type vanadia coating ceria nanodomains to achieve the synergetic modulation of lattice oxygen bulk diffusion and surface reaction. In the dehydrogenation step (reducer), ceria-vanadia redox catalysts donate lattice oxygen for the dehydrogenation of propane to produce propylene, $H_2O$, and $H_2$, affording a reduced valence state that can be reoxidized in the reoxidation step (oxidizer) by air to close the loop (Fig. 1a). Combining in situ spectroscopies, kinetics, and theoretical calculation, an intrinsically dynamic lattice oxygen "donator-acceptor" process is proposed, which accounts for the synergetic modulation of bulk diffusion and surface reaction in the core-shell redox catalyst. $O^{2-}$ generated from ceria oxygen carrier is boosted to diffuse and transfer to vanadia dehydrogenation sites via a concerted hopping pathway at the interface, stabilizing surface vanadia with moderate oxygen coverage without significant overoxidation or cracking.

## Results

### Formation of ceria-vanadia core-shell redox catalysts

The core-shell redox catalysts were prepared using a two-step incipient wetness impregnation method. The ceria-vanadia samples were named xV/yCeAl, where x(y) is the percent weight of V(Ce). The vanadia and ceria catalysts were obtained by $VO_x$ and $CeO_2$ supported on γ-$Al_2O_3$, respectively. At vanadia surface density of 4.3 V/$nm^2$ (Supplementary Table 1)[17–19] (6 V/30CeAl), atom-resolved high-angle annular dark-field scanning transmission electron microscope (HAADF-STEM) images identified vanadia mainly existed as monolayers and bilayers along ceria surface (Fig. 1b–d). Electron energy loss spectra (EELS) mappings of V $L_{2,3}$ and Ce $M_{4,5}$ edges affirmed vanadia sites anchored on widespread ceria nanodomains (Fig. 1e–h). The well-defined core-shell structure was further validated by line-scanning EELS that crossed an individual particle, wherein the outer shell was ~1 nm, corresponding to roughly two to three atomic vanadia layers (Fig. 1i). Judged from variations of V $L_{2,3}$ and O K edges, vanadia presented dominantly as a mixture of $V^{5+}$ and $V^{4+}$, while relative intensity ratios of Ce $M_{4,5}$ edges (1.11–1.19)[20] indicated the presence of $Ce^{3+}$ and $Ce^{4+}$ ((1), (2), (3) in Fig. 1b) inside one particle (Fig. 1j, Supplementary Fig. 2, and Supplementary Table 2)[21].

### Chemical looping oxidative dehydrogenation performance

Application of core-shell redox catalysts was proven in a continuous chemical looping oxidative dehydrogenation scheme (Supplementary Fig. 3). Ceria-vanadia redox catalysts exhibited traceable $CO_2$ (<3%) with high propylene selectivity of 93.5% and formation rate of 42.5 mmol $C_3H_6$/$g_{cat}$/h (at $5^{th}$ min in one cycle), implying excessive overoxidation or cracking were inhibited. At 600 °C and GHSV of 2500 $h^{-1}$, an average 90% propylene selectivity at propane conversion of 49% was obtained within 60 mins (Fig. 2b and Supplementary Fig. 4), superior to that of ceria (30CeAl) (78.3%), vanadia (6 V/Al) (71.6%), and state-of-the-art catalysts (Fig. 2c). Industrially relevant K-$CrO_x$/$Al_2O_3$[4,22] was compared under identical reaction conditions. The propylene space-time yield (STY) of ceria-vanadia redox catalysts was 10.3 mmol $C_3H_6$/$g_{cat}$/h, comparable to that of K-$CrO_x$/$Al_2O_3$ (10.6 mmol $C_3H_6$/$g_{cat}$/h) (Supplementary Fig. 3g). However, considering different reaction sites in two catalysts, propylene STY normalized by moles of V (13.3 mol $C_3H_6$/$mol_V$/h) was about five times higher than that normalized by moles of Cr (2.8 mol $C_3H_6$/$mol_{Cr}$/h). The deactivation rate constant ($k_d$) using a first-order deactivation model was used to determine its life in the

dehydrogenation step. Ceria-vanadia redox catalysts exhibited smaller $k_d$ (0.04 $h^{-1}$) than K-$CrO_x$/$Al_2O_3$ (0.99 $h^{-1}$) (Supplementary Figs. 5, 6). When the temperature was increased to 650 °C, propylene selectivity remained at 80%. However, pure vanadia showed quick deactivation ($k_d$ = 1.4 $h^{-1}$) and propylene selectivity decreased to 34%.

The reversible charge-discharge of lattice oxygen in ceria-vanadia redox catalysts was verified by in situ XRD. In the dehydrogenation step at 600 °C, diffract peaks of $CeO_2$ shifted to lower diffraction angles, e.g., the (111) diffract peak shifted from 28.4 ° to 28.0 ° due to the formation of larger $Ce^{3+}$ ions. Oxidation with air then recovered its position (Fig. 2a and Supplementary Fig. 7). During 300 long-term chemical looping cycles, structure durability and robust performance with an average 43.6% $C_3H_6$ yield and space-time yield of 9.9 mmol $C_3H_6$/$g_{cat}$/h was achieved (Fig. 2d and Supplementary Table 5). When altering either shell or core components in core-shell redox catalysts, comparable $C_3H_6$ formation rates were obtained (Supplementary Fig. 8). For chemical looping oxidative dehydrogenation of ethane, the ceria-vanadia redox catalysts also presented 92% ethylene selectivity with 31% ethane conversion at 600 °C (Supplementary Fig. 9), validating its potential application in the dehydrogenation of light alkanes. Compared with the commercialized Oleflex scheme (Supplementary Fig. 10 and Supplementary Tables 6–10), 45% of energy savings can be anticipated from the chemical looping oxidative dehydrogenation system (Fig. 2e), with separation being the main driver for energy consumption.

### Evidence of oxygen diffusion and surface reaction

When exposed to propane for 120 mins, peaks of $B_2$ and C in the Ce $L_3$-edge shifted to lower energy (Δ2.1 eV). The $B_0$ white line located at 5726 eV was then dominated, a characteristic of $Ce^{3+}$ (Fig. 3a), indicating the reduction of ceria ($Ce^{4+}→Ce^{3+}$) in the ceria-vanadia catalysts, in contrast to the negligible formation of $Ce^{3+}$ in pure ceria (Ce $L_3$-edge shift of Δ0.7 eV). V K pre-edge close to 5467 eV featuring $V^{4+}$ oxidation state kept nearly unmoved within 30 mins. After that, a decrease of pre-edge peak intensity and shift of edge position to lower energy (-Δ1.2 eV) occurred as the reduction of $CeO_2$ stopped (Fig. 3b and Supplementary Fig. 11). This implied that in the ceria-vanadia redox catalysts, vanadia tended to be reduced to lower valence states when oxygen was not timely supplied from ceria. For pure vanadia without $CeO_2$ supporting, V K pre-edge featuring $V^{5+}$ was easily and quickly reduced to $V^{3+}$ (V K-edge shift of 2.7 eV)[11]. Together, the changes of Ce $L_3$-edge and V K-edge indicated that ceria in ceria-vanadia redox catalysts acted as an "oxygen reservoir" that could supply the lattice oxygen to stabilize the surface vanadia, which accords with the previous research that ceria helped to oxidize the reduced vanadia via ceria lattice oxygen[23–25].

We further evidenced the dynamic evolution of lattice oxygen in ceria-vanadia redox catalysts. Raman spectra of $CeO_2$ were dominated by the strong $F_{2g}$ mode of fluorite phase at 464 $cm^{-1}$ with weak bands at 598 $cm^{-1}$ due to defect-induced (D) mode. With vanadia coating, in addition to V = O and V-O-V stretching, additional bands of V-O-Ce (859 and 720 $cm^{-1}$) emerged (Supplementary Fig. 1j, k), affirming the construction of vanadia-ceria interface[26,27]. Upon propane exposure, in situ Raman spectra verified the continuous reduction of $CeO_2$ in ceria-vanadia redox catalysts that the intensity of $F_{2g}$ mode dramatically decreased with the time on stream. It is noted that the band of V-O-Ce kept relatively stable. In contrast, intensity ratios of V-O-Ce band and $F_{2g}$ mode in terms of $I_{V-O-Ce}/I_{F_{2g}}$ increased, validating that Ce-O species in ceria were gradually consumed to supplement and stabilize interfacial and surface V-O species (Fig. 3c and Supplementary Fig. 12)[24,25]. As much oxygen was depleted after 30 min, the time was also shown in in situ XANES spectra. The $D_1$ band and G band corresponding with coke deposits, were then observed, implying that the cracking and coking of propane occurred on reduced vanadia sites. Comparatively, pure vanadia was readily reduced to $V^{3+}$, which leads to more coke deposition in the characteristic of the more dominated intensity of $D_1$ and G band[27–29].

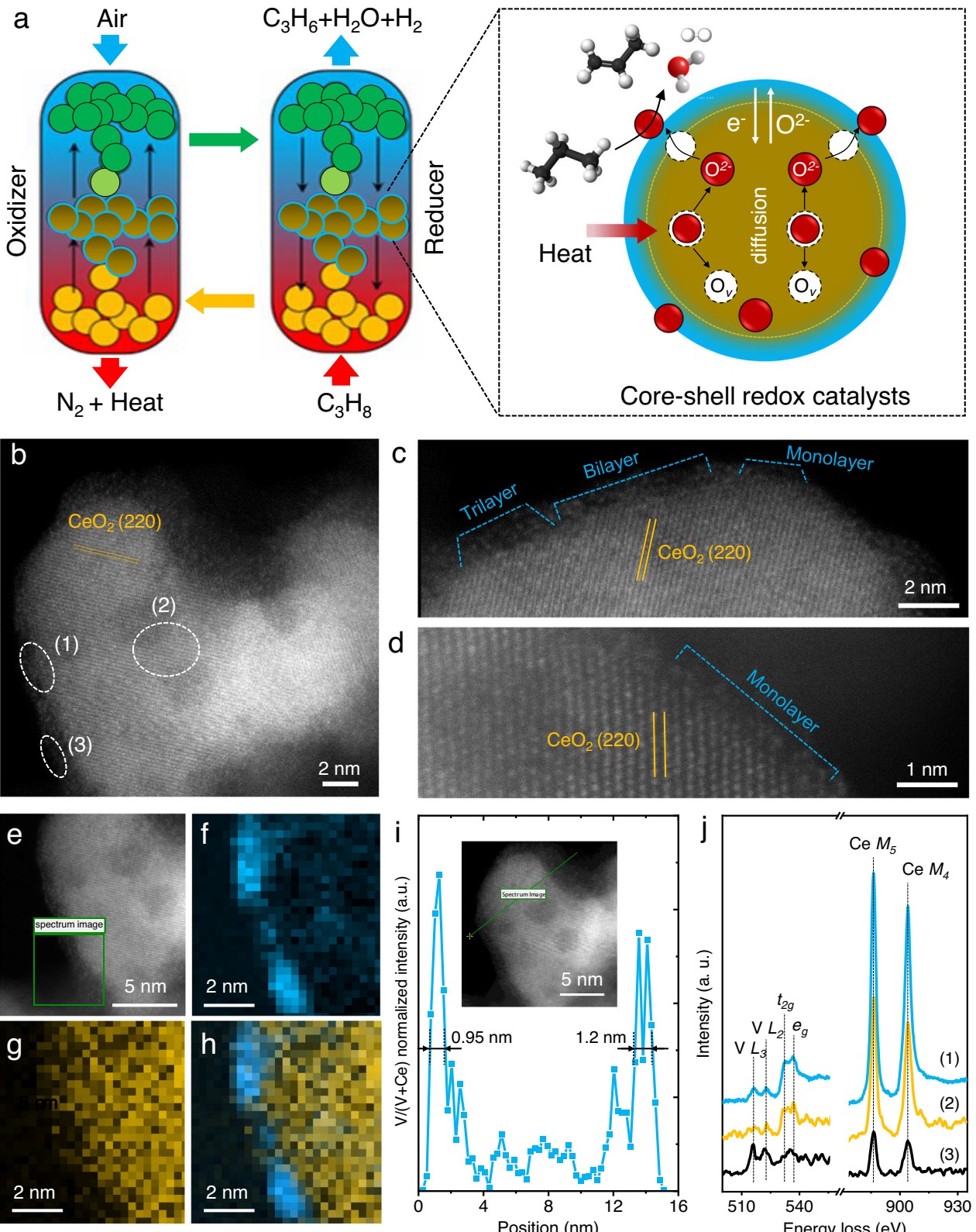

**Fig. 1 | Identification of vanadia layers that coat ceria nanodomains. a** Diagram of core-shell redox catalysts in propane dehydrogenation by the chemical looping engineering: dehydrogenation and oxidation in fuel reactor (reducer) and air reactor (oxidizer), respectively. **b−d** HAADF-STEM images and **e−h** EELS mappings of core-shell ceria-vanadia redox catalysts (6 V/30CeAl): (**f**): V; (**g**): Ce; (**h**): V + Ce. **i** Line-scanning EELS. **j** EELS of the domains ((1), (2), (3)) in (**b**).

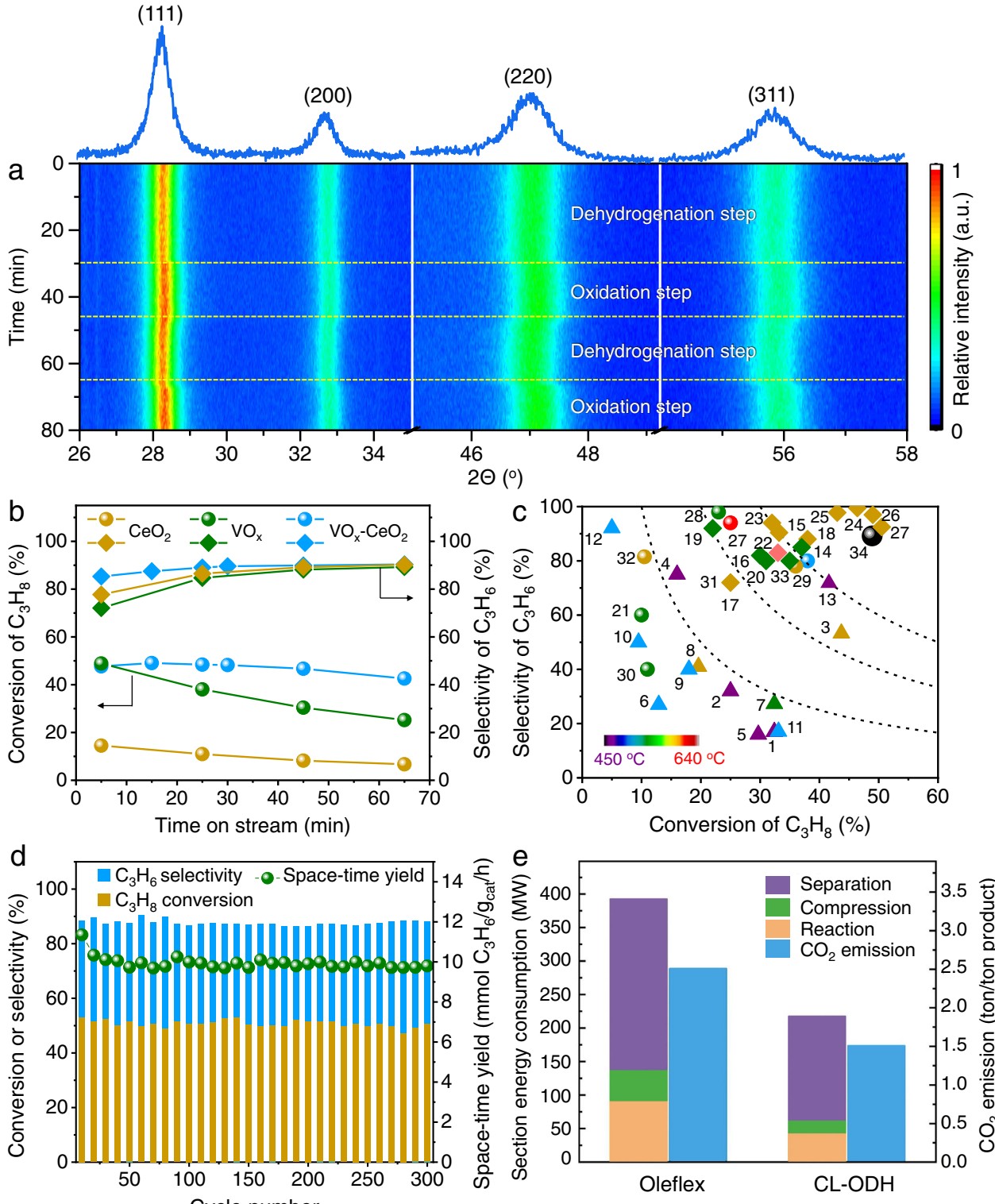

**Fig. 2 | Chemical looping oxidative dehydrogenation performance. a** In situ XRD patterns of ceria-vanadia redox catalysts (6 V/30CeAl). **b** Comparison of ceria (30CeAl), vanadia (6 V/Al), and ceria-vanadia redox catalysts (6 V/30CeAl). Conditions: 600 °C, GHSV = 2500 h⁻¹, C₃H₈/N₂ = 0.25. **c** Comparing ceria-vanadia redox catalysts (6 V/30CeAl) with established oxide-based and Pt-containing catalysts (see Supplementary Tables 3, 4). Motifs of triangle, rhombus, and sphere represent ODH, PDH, and CL-ODH, respectively. **d** Cyclic performance over ceria-vanadia redox catalysts (6 V/30CeAl). Dehydrogenation step: 600 °C, GHSV = 2500 h⁻¹, C₃H₈/N₂ = 0.25 for 30 min; Inert purge: 600 °C, N₂ = 40 mL/min for 5 min; Oxidation step: 600 °C, 20 vol.% O₂/N₂ = 20 mL/min for 15 min. **e** Comparison of energy consumption and CO₂ emission of traditional Oleflex technology and chemical looping scheme (see Methods and Supplementary Tables 6–8).

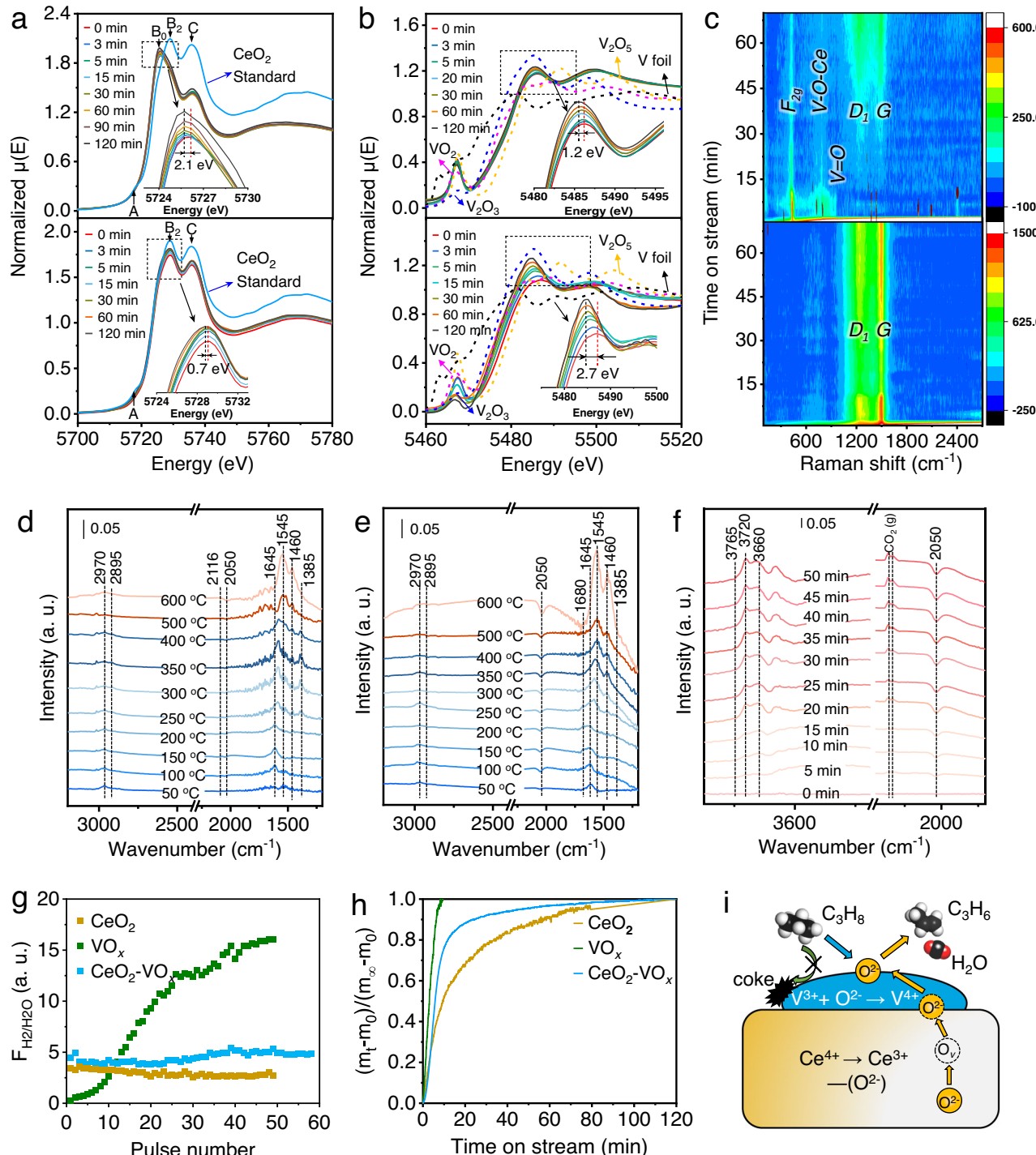

**Fig. 3 | Experimental evidence of oxygen diffusion and surface reaction. a** In situ XANES spectra of Ce $L_3$-edge (CeO$_2$ standards as the references) over ceria-vanadia (6 V/30CeAl) (top) and pure ceria (30CeAl) (bottom) and **b** V K-edge (V foil, V$_2$O$_5$, VO$_2$, and V$_2$O$_3$ standards as the references) over ceria-vanadia (6 V/30CeAl) (top) and pure vanadia (6 V/Al) (bottom) at 600 °C under the flow of 20% C$_3$H$_8$/N$_2$ (20 mL/min). **c** In situ Raman spectra of ceria-vanadia (6 V/30CeAl) (top) and vanadia (6 V/Al) (bottom) at 600 °C under the flow of 20% C$_3$H$_8$/N$_2$ (20 mL/min). In

situ DRIFTS spectra of temperature-programmed and isothermal propane dehydrogenation over ceria-vanadia (6 V/30CeAl) (**d**) and vanadia (6 V/Al) (**e, f**). **g** The calculated ratios of H$_2$/H$_2$O during the C$_3$H$_8$ transient pulses at 600 °C. **h** Experimental relaxation curve in the form of fractional weight change as a function of time at 600 °C under the flow of 20% C$_3$H$_8$/He (10 mL/min). **i** Schematic representation of the concerted oxygen diffusion in the ceria-vanadia redox catalyst. The black, white, and red spheres represent C, H, and O atoms.

In situ diffuse reflectance infrared Fourier transform spectroscopy (DRIFTS) upon propane exposure identified the co-existence of dehydrogenation and cracking of propane induced by this dynamic oxygen evolution. Peaks ascribed to asymmetric and symmetric CH$_3$ stretching modes (2970 and 2875 cm$^{-1}$) started at 100–150 °C (Fig. 3d)[30]. The presence of a band centering at 1645 cm$^{-1}$

(v(CH$_3$CH=CH$_2$)) implied that the propyl complex was oxidatively dehydrogenated to propenyl by heterolytically subtracting H to neighboring V-O sites, leading to the occurrence of vanadium hydroxyl band (V-OH, 3660 cm$^{-1}$)[28]. However, a peak of v(C=O) (1680 cm$^{-1}$) attributed to acetone, the intermediate of overoxidation of propane to CO$_x$, was dominated on the pure VO$_x$ catalysts when the temperature

was higher than 150 °C, along with a significantly negative V = O band induced by the ready reduction of vanadia ($V^{5+} \rightarrow V^{3+}$) (Fig. 3e, f and Supplementary Fig. 11)[11,28]. This "over quick" oxygen removal would induce the transformation of oxidative to non-oxidative dehydrogenation and the occurrence of propane cracking. At 250–600 °C, two peaks at 1545 and 1460 cm$^{-1}$ attributed to the unsaturated or aromatic species, the precursors of coke deposits[27,28] that lead to fast deactivation were observed, which were also evidenced by the more dominated $D_l$ and $G$ band in situ Raman spectra on pure vanadia catalysts during dehydrogenation step.

As evidenced by the in situ spectroscopies, the release of lattice oxygen would induce the existence and transformation of different reaction periods, including overoxidation, oxidative dehydrogenation, and non-oxidative dehydrogenation. Under differential reactor operation by controlling the propane conversion lower than 10%, the $C_3H_6$ formation rate showed a linear relationship with $C_3H_8$ pressure, while $C_3H_8$ conversion kept identical at different $C_3H_8$ pressures, indicating the rate of propene formation is typically related to propane partial pressure, i.e., a first-order reaction with respect to propane (Supplementary Fig. 13). To clarify the contribution of oxidative and non-oxidative dehydrogenation, the formation of $H_2O$ and $H_2$ over ceria-vanadia redox catalysts in their dehydrogenation tests were investigated. As shown in Supplementary Fig. 13, the initial ratio of $H_2O$ to $H_2$ at the 5$^{th}$ min was 0.44; however, it decreased to about 0.05 after 60 mins. Therefore, oxidative dehydrogenation could be more dominated in the initial period (less than 5 mins) and it was transformed to non-oxidative dehydrogenation with time, accounting for the introduction of the reoxidation step to recover the lattice oxygen after the 30-min dehydrogenation test during the continuous dehydrogenation-reoxidation cycles with the ratios of $H_2O$ to $H_2$ of ~0.21.

## Transient pulses and oxygen release kinetics

To catch the transient distributions of products, $C_3H_6$, $CO_x$, $H_2O$, and $H_2$, especially in the initial period, $C_3H_8$ transient pulse experiments were employed at 600 °C using online mass spectrometry (MS) (Supplementary Fig. 14). $CO_x$ was firstly observed due to the overoxidation on the active surface oxygen species (period I), then it would undergo selectively oxidative dehydrogenation to $C_3H_6$ and $H_2O$ caused by lattice oxygen (period II). After the lattice oxygen were fully released, cracking or hydrogenolysis occurred, which was more dominated for pure vanadia catalysts, leading to the formation of $CH_4$ (period III). Intensity ratios of $H_2/H_2O$ defined as $F_{H2/H2O}$, a sign of H combustion, maintained at five over ceria-vanadia redox catalysts (Fig. 3g), implying that $O^{2-}$ diffusion from ceria oxygen carrier leads to a pseudo-steady-state H combustion at surface vanadia sites[6,31]. In contrast, pure vanadia showed stage-divided products, along with the tenfold increased intensity ratios of $H_2/H_2O$ with pulses of $C_3H_8$, indicating the presence of $O^{2-}$ gradient and transport limitation in the pure vanadia[31]. Gradually decreased ratios of $H_2/H_2O$ in pure ceria implied surface H abstraction were sluggish due to the less active C–H dissociation center of $CeO_2$. We then designed a temperature-programmed surface reaction (TPSR) to explore the surface reaction route. Initial dehydrogenation temperatures of $C_3H_8$ on pure vanadia and ceria-vanadia redox catalysts were at 338 and 340 °C, respectively, lower than that of pure ceria (412 °C) (Supplementary Fig. 14), indicating the presence of low-temperature C–H dissociation V-O centers ascribed from vanadia catalysts, corresponding to increased medium acidic sites as shown in $NH_3$-TPD profiles (Supplementary Fig. 15).

Oxygen release kinetics provide further quantified rates of oxygen diffusion and surface reaction. 1.83 wt% active oxygen in ceria-vanadia redox catalysts was continuously removed during the dehydrogenation step (Supplementary Figs. 16, 17), much higher than that of the ceria oxygen carrier (0.37 wt%). However, quick oxygen removal (0.16 wt%) and coke accumulation were observed over the vanadia catalyst. Derived from thermogravimetry relaxation and diffusion-

reaction equations (Fig. 3h)[32], bulk diffusion coefficients ($D_{diff}$) and surface exchange coefficient ($k_{chem}$) of ceria-vanadia redox catalysts at 550–600 °C were close to one order of magnitude higher than that of ceria oxygen carrier (Supplementary Table 11), validating the dramatic acceleration of oxygen diffusion by ceria–vanadia interaction[23,25,33]. Derived from Arrhenius plots of $D_{diff}$ and $k_{chem}$ at 550–625 °C, ceria-vanadia redox catalysts possessed both lower activation energy ($E_a$) of oxygen migration (131.7 vs. 179.9 kJ/mol for ceria) and surface exchange barrier (89.8 vs. 176.3 kJ/mol for ceria) (Supplementary Fig. 18), which accounts for the modulation of oxygen diffusion and surface reaction to produce propylene, as shown in transient pulse experiments.

## DFT calculations on oxygen diffusion and surface reaction

Atomic-level details of oxygen diffusion and surface reaction over ceria-vanadia redox catalysts were investigated by density functional theory (DFT) calculations. With the elimination of O of $VO_x$ (O(V)) in $CeO_2$-$VO_x$ (Supplementary Fig. 19), the outermost O of $CeO_2$ (O(Ce)) started to coordinate with V to form V-O-Ce interface that exposes $Ce^{3+}$ centers (Fig. 4a). Bader charge analysis further proves electrons accumulate on Ce with reduction of $VO_x$ (Supplementary Fig. 20). Meantime, $VO_x$ preserves its valency state until ML-$V_2O_3$ period (Supplementary Table 12). Existence of monolayer-$V_2O_5$ (ML-$V_2O_5$) on $CeO_2$ is expectedly activated surface oxygen with lower oxygen vacancy ($O_{vac}$) formation energies than that of pure $CeO_2$. When surface oxygen was entirely removed, and ML-$VO_2$ formed, oxygen diffusion from $CeO_2$ to $VO_x$ turned endothermic with a reaction energy of 0.62 eV. However, it became exothermic with further oxygen release in $VO_x$ between ML-$VO_2$ and ML-$V_2O_3$, verifying the continuous oxygen transfer from O(Ce) to O(V) sites (Supplementary Fig. 21). Derived from the structure of ML-$VO_2$ on $CeO_2$, a concerted hopping pathway was preferred that $O^{2-}$ diffusion from O(Ce) (1.17 eV) and subsequential transfer from interface V-O-Ce to V=O mediated by bridge V-O-V exhibited the lowest barrier of 0.95 eV (Fig. 4b) as shown in in situ Raman spectra, lower than that of direct hopping pathway (1.45 eV) (Fig. 4c) and isolated hoping pathway (1.47 eV) (Supplementary Fig. 22a). As a result, ML-$VO_2$ on $CeO_2$ could be stabilized by the intrinsically dynamic "donator-acceptor" process. Four V atoms were connected to form a ring-like structure at the interface, leading to an atomic ratio of V/O at 1:2, evidenced by in situ V K-edge QXANES. For each [$V_4O_8$] unit (Supplementary Fig. 23), two terminal V=O bonds were kept, which would not exist in crystalline $VO_2$ (Supplementary Fig. 22b, c)[11,34].

Valence states of V have been believed to manipulate the propylene selectivity[35,36]. The following calculations on the formation pathways of $H_2O$(g) and $H_2$(g) showed that ML-$VO_2$ has a strong preference to proceed oxidative dehydrogenation compared with ML-$V_2O_3$ (Supplementary Fig. 24) and exhibited lower barriers of first and second dehydrogenation of propane to propylene, but higher barriers for acetone formation, a significant intermediate of overoxidation, than crystalline $V_2O_5$ and ML-$V_2O_3$ (Fig. 4d and Supplementary Fig. 25), according to experimental results that ceria-vanaida redox catalysts showed higher initial propane conversion and propylene selectivity than 6 V/Al and crystalline $V_2O_5$. These results firmly support that lattice oxygen transfer from ceria was to stabilize moderate V valence states and oxygen coverage for selective dehydrogenation without significant overoxidation or cracking and coking.

In conclusion, the ceria-vanadia core-shell redox catalysts have been designed for propane dehydrogenation via chemical looping engineering, which exhibited the highest 93.5% propylene selectivity and 43.6% propylene yield during the long-term dehydrogenation-oxidation cycles. An intrinsically dynamic lattice oxygen "donator-acceptor" process in core-shell redox catalyst was proposed by the combination of in situ XAS, Raman, transient pulses and oxygen release kinetic analysis, that the lattice oxygen generated from ceria

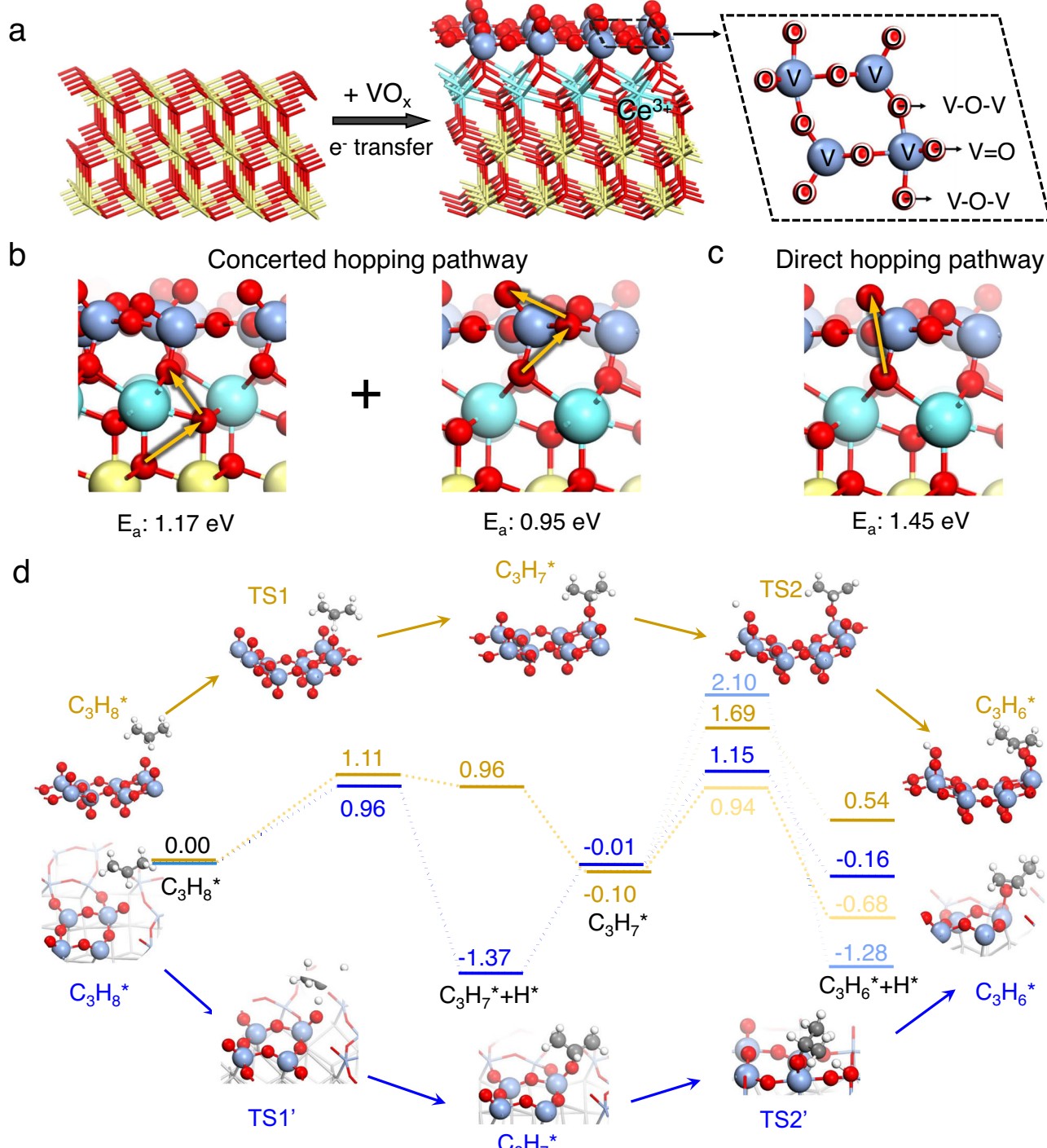

**Fig. 4 | DFT calculations on oxygen diffusion and surface reaction. a** Models of CeO$_2$ and ML-VO$_2$ (inset [V$_4$O$_8$] units). **b** Optimal concerted hopping pathway in ML-VO$_2$ contrast to **c** direct hopping pathway. E$_a$ represents the energy barrier. **d** The calculated energy profiles on ML-VO$_2$ (blue) and crystalline V$_2$O$_5$ (yellow). Reaction steps include (i) dehydrogenation from propane to absorbed propyl, (ii) dehydrogenation from absorbed propyl to absorbed propene, and (iii) dehydrogenation from absorbed propyl to acetone. TS represents the transition states. V: dark blue; Ce; bright blue; O: red; C: black; H: white.

were boosted to diffuse (close to one order of magnitude higher $D_{diff}$ at 550–600 °C that that of ceria oxygen carrier), and transfer to surface vanadia dehydrogenation sites via a concerted hopping pathway at the interface, stabilizing surface vanadia for selective dehydrogenation without significant overoxidation or cracking. Atomic-level details of oxygen diffusion and surface reaction over ceria-vanadia redox catalysts were further revealed by DFT calculations that derived from the structure of ML-VO$_2$ on CeO$_2$, O$^{2-}$ diffusion from O(Ce) and subsequent transfer from interface V-O-Ce to V=O mediated by bridge V-O-V exhibited the lowest barrier lower than that of the direct and isolated hoping pathway. This study opens the possibilities for the design and application of efficient core-shell redox catalysts in selective oxidations and chemical looping systems.

## Methods

### Material preparation

The VO$_x$-CeO$_2$ catalysts were prepared via two-step-impregnation procedure by dissolving Ce(NO$_3$)$_4$.6H$_2$O (99.5%, Aladdin (China)

Chemical Co., Ltd) in deionized water (2 mL/g $Al_2O_3$) with γ-$Al_2O_3$ (Adamas, 99.99%, $S_{BET}$ = 180 m$^2$/g) as support to well disperse the cerium oxides. Then $VO_x$ was impregnated by dissolving $NH_4VO_3$ (99.0%, Tianjin Guangfu Technology Development Co. Ltd.) and oxalic acid (99.0%, Aladdin Industrial Corporation) [$NH_4VO_3$/oxalic acid = 0.5 (mole ratio)] in deionized water. The samples were dried at 80 °C for 12 h and then calcinated at 600 °C for 3 h. The $VO_x$-$CeO_2$ samples were named xV/yCeAl, where x(y) is the percent weight ratio of V(Ce). The vanadia and ceria catalysts were obtained by $VO_x$, and $CeO_2$ supported on γ-$Al_2O_3$ with the same V and Ce loadings with $VO_x$-$CeO_2$, respectively, as reference samples. An industrially relevant K−$CrO_x$/$Al_2O_3$ catalyst was carried out (mimic Catofin from Lummus) to compare the $VO_x$-$CeO_2$ catalyst. For K-$CrO_x$/$Al_2O_3$ reference catalysts, 20 wt% Cr and 1.0 wt% K were utilized with $Cr(NO_3)_3 \cdot 9H_2O$ (99.5%, Aladdin (China) Chemical Co., Ltd)[37]. The catalysts were dried for 2 h at 60 °C and overnight at 120 °C, followed by calcination at 600 °C for 3 h. The amount of Cr and K was about 19.8 wt% and 0.98 wt% based on the weight ratio of M to $Al_2O_3$, which were determined by inductively coupled plasma optical emission spectroscopy (ICP-OES),

## Reaction test

Reactivity tests were performed in a quartz fixed-bed reactor (8 mm ID) loaded with 0.5 g catalysts (20–40 mesh) mixed with 1 mL of quartz particles at atmospheric pressure. Switching between propane and air flows was employed during tests. The bed temperature was typically 600 °C, and the samples were reduced using propane (4 mL/min) diluted in nitrogen (16 mL/min) at 1.4 atm for 30 min. The samples were then reoxidized using air (20 mL/min) for 15 min. During the reduction and reoxidation reaction period, an inert period (40 mL/min of nitrogen) of about 5 min was inserted to prevent the mixing between propane and air. One redox cycle was thus completed. Afterward, the second reaction cycle was started by switching between propane and air flows. Exhaust streams were analyzed using an online GC (2060) equipped with a flame ionization detector (Chromosorb 102 column) and a thermal conductivity detector ($Al_2O_3$ Plot column). In addition, high-time resolution measurements were performed by Agilent 490 Micro GC equipped with three channels (MS 5A Plot column, PoraPlot Q, and PoraPlot Q), which can quickly quantitatively analyze the product compositions about once a minute.

The instantaneous propane conversion and propylene selectivity based on all products (including coking formation) and gas phase products are defined as the instantaneous values at the different time on stream, according to Eq. (1) and Eq. (2) (2a: selectivity including coke formation and 2b: gas selectivity). The propylene yield was calculated based on propane conversion and propylene selectivity (including coke formation):

$$Con (\%) = 100 \times ([F_{C3H8}]_{inlet} - [F_{C3H8}]_{outlet})/[F_{C3H8}]_{inlet} \quad (1)$$

$$Sel (\%) = 100 \times [F_{C3H6}]_{outlet}/([F_{C3H8}]_{inlet} - [F_{C3H8}]_{outlet}) \quad (2a)$$

$$Sel (\%) = 100 \times 3 \times [F_{C3H6}]_{outlet}/\left(\sum_{ni} \times [Fi]_{outlet}\right) \quad (2b)$$

$$Yield (\%) = Con (\%) \times Sel (\%)/100 \quad (3)$$

The average conversion and selectivity within $t$ minutes in the dehydrogenation stage were defined as the integral conversion and selectivity in Eq. (4) and Eq. (5) dividing $t$ minutes during the dehydrogenation stage.

$$Con_{int} (\%) = \left(\int Con (\%) \, dt\right)/t \quad (4)$$

$$Sel_{int} (\%) = \left(\int Sel (\%) \, dt\right)/t \quad (5)$$

The propylene formation rate was defined according to Eq. (6).

$$Rate = [F_{C3H6}]_{outlet}/m \quad (6)$$

A first-order deactivation model was used to evaluate the catalyst stability:

$$k_d = (\ln[(1 - X_{final})/X_{final}] - \ln[(1 - X_{initial})/X_{initial}])/t \quad (7)$$

Where $i$ stands for different side products in exhaust gases, $n_i$ is the number of carbon atoms of side products $i$, and $F_i$ is the corresponding molar flow rate (mol/h). [$F_{C3H6}$]$_{outlet}$ is the flow of propylene out of the reactor (mol/h). [$F_{C3H8}$]$_{outlet}$ is the flow of propane out of the reactor (mol/h). [$F_{C3H8}$]$_{inlet}$ is the flow of propane in of reactor (mol/h). t is the time during the dehydrogenation stage (min). $m$ is the weight of catalysts ($g_{cat}$). $X_{initial}$ and $X_{final}$, respectively, represent the conversion measured at the initial and final period of an experiment. $t$ represents the reaction time (h). $k_d$ is the deactivation rate constant (h$^{-1}$). A high $k_d$ value means rapid deactivation, that is, low stability.

As internal standard experiments indicated that no significant coking or tar formation occurred under the conditions tested, a mass balance was used to calculate the yields. The selectivity and conversion for carbonaceous species were calculated relative to the carbon mass balance. $H_2O$ was calculated from a hydrogen balance. Molar flows of propane and reaction products are determined by mass flow controllers and GC. Flowrates of propane (industrial grade) and nitrogen (UHP) were controlled using two mass flow controllers (Bronkhorst) and calibrated to each individual gas to allow total flow rates of 0–20 mL min$^{-1}$ and 10–50 mL min$^{-1}$. Exhaust streams were analyzed using an online GC (2060) equipped with a flame ionization detector (Chromosorb 102 column) and a thermal conductivity detector ($Al_2O_3$ Plot column), and Agilent 490 Micro GC equipped with three channels (MS 5A Plot column, PoraPlot Q and PoraPlot Q).

## Characterizations

X-ray powder diffraction (XRD) patterns were performed with 2θ values between 10° and 80° by using a Rigaku C/max-2500 diffractometer with the graphite filtered Cu Kα radiation (λ = 1.5406 Å), operated at 40 mA and 40 kV.

Raman spectra were recorded using a Renishaw inVia reflex Raman spectrometer with a 532 nm Ar ion laser beam. Samples were pretreated at 300 °C for 2 h to eliminate the presence of hydrated species.

Transmission electron microscope (TEM) was carried out on a JEM-2100F transmission electron microscope under a working voltage of 200 kV. The aberration-corrected scanning transmission electron microscopy (AC-STEM) images and EELS spectra were characterized on FEI Titan Cubed Themis G2 300 (FEI) 200 kV, capable of sub-angstrom resolution at Tianjin University of Technology. The sample powder was dispersed in deionized water by ultrasonic and deposited on a copper grid coated with an ultrathin holey carbon film.

XPS measurements were taken on a PHI 1600 ESCA instrument (PE Company) equipped with an Al Kα X-ray radiation source (h$v$ = 1486.6 eV). Before measurements, all the samples were reduced under a flow of $H_2$ at 600 °C for 1 h. The binding energies were calibrated using the C 1$s$ peak at 284.5 eV as a reference.

$H_2$ temperature-programmed reduction ($H_2$-TPR) was performed on the chemisorption apparatus (Micromeritics AutoChem II 2920). Typically, 100 mg samples were pretreated at 400 °C for 1 h in an Ar stream and then cooled to 80 °C. The analysis was carried out in a

mixture of 10 vol% $H_2$ in Ar (30 mL/min), ramping temperature from 80 °C to 800 °C at 10 °C/min.

The $C_3H_8$ pulse experiment was measured on a Micromeritics Autochem II 2920 instrument equipped with a Hiden QIC-20 mass spectrometer. About 200 mg samples were pretreated at 400 °C for 1 h and raised up to 600 °C in the Ar stream. The analysis was carried out in a mixture of propane (5 mL/min) at 600 °C for 60 min. The output products ($C_3H_8$, $C_3H_6$, $CO_2$, $CH_4$, $H_2$, and $H_2O$, m/e equals 29, 41, 44, 16, 2, and 18, respectively) were measured via mass spectrometer.

The thermogravimetric relaxation experiment was performed on Themys. About 30 mg samples were pretreated at 300 °C for 2 h under He (100 mL/min) to eliminate the presence of hydrated species and then reacted in a mixture of 20% $C_3H_8$/He (10 mL/min) at 550, 575, 600, and 625 °C, respectively. The general way to model this process is by solving the diffusion coefficient ($D_{diff}$), and surface exchange coefficient ($k_{chem}$) contained diffusion equation. The mathematical processing can be described as follow:

The diffusion can be seen only in the radius direction of a sphere, and the diffusion coefficient $D_{diff}$ remains constant[32,38].

$$\frac{\partial C_O}{\partial t} = D_{diff}\left(\frac{\partial^2 C_O}{\partial r^2} + \frac{2}{r}\frac{\partial C_O}{\partial r}\right) \tag{8}$$

where $C_O$ is the oxygen species concentration in the catalysts, $t$ is the reaction time; $r$ is the radius at the specific position of the catalysts powder.

The oxygen exchange happens at the gas/solid interface, and the process is deemed a first-order reaction.

$$D_{diff}\frac{\partial C_O}{\partial r} = k_{chem}[C_O(\infty) - C_O(t)], r = R \tag{9}$$

where $k$ is the surface exchange coefficient; $C_O(\infty)$ is the oxygen species concentration at infinite reaction time; $C_O(t)$ is the oxygen species concentration at reaction time $t$; $R$ is the average radius of the catalyst powder.

The reaction proceeds thoroughly under infinite time.

$$C_O = C_O(0), t = 0 \tag{10}$$

$$C_O = C_O(\infty), t = \infty \tag{11}$$

By solving PDE under the above conditions, the temporal and spatial distribution of oxygen concentration can be acquired in the following formula:

$$\frac{C_o(t,r) - C_o(\infty)}{C_o(0) - C_o(\infty)} = \frac{2LR}{r}\sum_{n=1}^{\infty}\frac{e^{\left(-D_{diff}\frac{\beta_n^2 t}{R^2}\right)}}{\left[\beta_n^2 + L^2 - L\right]} \cdot \frac{\sin\left(\beta_n\frac{r}{R}\right)}{\sin(\beta_n)} \tag{12}$$

with

$$\beta_n\cot(\beta_n) + L - 1 = 0 \tag{13}$$

$$L = \frac{Rk_{chem}}{D_{diff}} \tag{14}$$

where $C_O(\infty)$ is the oxygen species concentration at infinite reaction time; $C_O(t, r)$ is the oxygen species concentration at reaction time $t$; $C_O$ is the oxygen species concentration in the catalysts, $t$ is the reaction time; $r$ is the radius at the specific position of the catalysts powder; $R$ is the average radius of the catalyst powder. $\beta_n$ is a dimensionless parameter. L is the characteristic length of the solid one; $k_{chem}$ is the surface exchange coefficient; $D_{diff}$ is the diffusion coefficient.

The temporal and spatial distribution of oxygen concentration can be transformed to the temporal distribution of catalyst powder weight by a spherical integral of the oxygen concentration distribution.

$$\frac{m(t) - m(0)}{m(\infty) - m(0)} = 1 - \sum_{n=1}^{\infty}\frac{6L^2 e^{\left(-D_{diff}\frac{\beta_n^2 t}{R^2}\right)}}{\beta_n^2\left[\beta_n^2 + L^2 - L\right]} \tag{15}$$

where $m(0)$ is the weight of the catalysts at reaction time 0; $m(t)$ is the weight of the catalysts at reaction time $t$; $m(\infty)$ is the weight of the catalysts at infinite reaction time.

The TG data can be fitted with the weight distribution formula to acquire the $D_{diff}$ and $k_{chem}$. The fitting is completed with MATLAB® software with the plotting algorithm in the third part toolbox OPTI-master. All data are checked with correlations and confidence intervals to prevent over-parameterization and ensure credibility.

In situ DFIRTS of the catalysts were obtained by Nicolet 6700 spectrometer with a stainless-steel cell connected to a gas-dosing and evacuation system. About 30 mg of the catalysts was compressed into in situ cell equipped with ZnSe windows. The IR spectra can be collected with a resolution of 8 cm⁻¹.

In situ Raman spectra were recorded using a Renishaw inVia reflex Raman spectrometer with a 532 nm Ar ion laser beam. Samples were pretreated at 300 °C for 2 h to eliminate the presence of hydrated species and then reacted at 600 °C under the flow of 20% $C_3H_8$/$N_2$ (20 mL/min).

In situ XRD patterns were recorded using SmartLab equipped with an in situ cell and recorded in the 2θ range of 10–80°. The sample was reduced using 20% $C_3H_8$/$N_2$ (20 mL/min) at 600 °C for 60 min. The samples were then reoxidized using air (20 mL/min) for 30 min. During the reduction and reoxidation reaction period, an inert period (40 mL/min of nitrogen) of about 10 min was inserted to prevent the mixing between propane and air. One redox cycle was thus completed. Afterward, the second reaction cycle was started by switching between propane and air flows.

In situ X-ray absorption near-edge structure (QXANES) was performed at the XAFS station in the 1W1B beamline of Beijing Synchrotron Radiation Facility (BSRF). Each spectrum was collected under the normal mode and the sampling time was 3 min. During the in situ experiments, the sample was reduced using 20% $C_3H_8$/$N_2$ (20 mL/min) at 600 °C, and a V foil was employed for energy calibration. The XANES spectra of metallic V (V foil), $V_2O_5$, $VO_2$, and $V_2O_3$ were employed as the references. All the data were analyzed by the software of Athena.

## Computational details

All the spin-polarized DFT calculation was performed by using the Vienna ab initio simulation package (VASP, 5.4.4 version[39]), employing the projector augmented-wave model. To depict the exchange-correlation effect, we employed the Perdew, Burke, and Ernzerhof (PBE) functional within the generalized gradient approximation (GGA)[40]. In order to correct the on-site Coulomb correlation of occupied Ce 4$f$ and V 3$d$ orbitals, the Hubbard U corrections were adopted for both Ce and V elements, using the model proposed by ref. 41. The $U_{eff}$ ($U_{eff}$ = Coulomb (U) – exchange (J)) values are 4.5 eV for Ce and 3.2 eV for V[42]. The valence wave functions were expanded by plane wave with a cutoff energy of 400 eV. J. Sauer et al. investigated the complex $VO_x$-$CeO_2$ system using their global optimization algorithm[42]. We conducted our calculation based on the optimal structures of 4 × 4 three layers ML - $V_2O_5$ and ML - $VO_2$ they found. The Brillouin zone was sampled using only the zone center Γ for geometry optimization and self-consistent calculations. All the structures were optimized until the force on each atom was less than 0.02 eV Å⁻¹. The relaxation of the electronic degrees of freedom will be stopped if the total energy change between two steps is smaller than $10^{-4}$ eV.

$H_2(g)$ and $H_2O(g)$ were used as a reference for the calculation of oxygen vacancy formation energy:

$$\triangle E_v = E_{\text{surface with Ov}} + H_2O(g) - E_{\text{clean surface}} - H_2(g) \tag{16}$$

In potential energy diagrams, the energy of $C_3H_8$ in the gas phase is taken as a reference, and the energy of intermediates is corrected with $H_2$ in the gas phase. The adsorption energy is defined as:

$$\triangle E_{ads} = E_{\text{surface} + C_3H_x} + (4 - 0.5x)E_{H_2(g)} - E_{\text{clean surface}} - E_{C_3H_8(g)} \tag{17}$$

Moreover, all barrier calculations were performed by using the climbing-image nudged elastic band method (NEB)[43] and further improved by the dimer method[44]. The activation barrier $E_a$ was calculated based on the following equation:

$$\triangle E_a = E_{TS} - E_{IS} \tag{18}$$

## Process simulation

Process simulation was performed with ASPEN Plus. A pair of reactors is applied for the chemical looping oxidative dehydrogenation (CL-ODH) process, of which one is the oxidative dehydrogenation reactor, and the other is the regenerator. Traditional propane dehydrogenation (PDH) process (Oleflex technology) is operated Oleflex process makes use of four adiabatic moving-bed reactors in series to convert paraffin into olefin, which is followed by a continuous catalyst regenerator, and the reaction heat is supplied by interstage reheating furnace[1,45]. Downstream includes quenching, compression, deep freezing, PSA, Light/heavy hydrocarbon separation, and P-P splitter units. In a chemical looping scheme, the reaction and regeneration of redox catalyst are isolated spatially or temporally. Propane is converted to propane in a dehydrogenation reactor. Meanwhile, the deactivated catalyst circulates between the reactor and regenerator by means of transfer lines[9,46]. Heat supply from the regeneration and coke combustion, as well as additional fuel gas. Downstream includes $CO_2$ generation for the introduction of lattice oxygen from an additional $CO_2$ removal unit (amine absorption process). The processes are calculated with the feed of 37500 kg/hr (96 wt.% propane, 2 wt% ethane, 2 wt% butane).

## Data availability

The data supporting the findings of the study are available within the paper and its Supplementary Information[47]. Source data are provided with this paper.

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

## Acknowledgements

This work is supported by the National Key R&D Program of China (2021YFA1501302), the National Science Foundation of China (Nos. 22121004, 22122808, and 22108201), the China Postdoctoral Science Foundation (No. BX2021212 and 2021M692383), the Haihe Laboratory of Sustainable Chemical Transformations, the Program of Introducing Talents of Discipline to Universities (BP0618007), and the XPLORER PRIZE for financial support.

## Author contributions

J.G. conceived and coordinated the research. S.C. and Y.X. contributed to catalyst synthesis and catalytic experiments. S.C. and Z.L. performed the XANES measurements and analyzed the data. R.L. and Z.-J.Z. carried out DFT calculations. H.S. contributed to the oxygen release kinetic fitting. C.Z. performed the Aspen Plus simulation. S.C., R.L., Z.-J.Z., C.P., and J.G. wrote the manuscript. All authors participated in the discussion of the research.

## Competing interests

The authors declare no competing interests.
