## [Peer Review File · Nature Communications]

Concerted Oxygen Diffusion Across Heterogeneous Oxide Interfaces for Intensified Propane DehydrogenationREVIEWER COMMENTS

Reviewer #1 (Remarks to the Author):

Conversion of available but chemically inert short alkanes into value-added chemicals is an important topic both from fundamental and application viewpoints. The present manuscript reports about propane dehydrogenation to propene over V-Ce-containing catalysts under O₂-free conditions. Such operation mode is called as chemical looping approach, according to which the reaction proceeds oxidatively with participation of catalyst lattice oxygen. The group of Jinlong Gong is well known for the activity in this area. In this study, the authors prepared core(CeO₂)-shell(V₂O₅) catalysts based on Al₂O₃ support. The idea is to tune activity and selectivity of V₂O₅ through the redox properties of ceria. A similar idea is used for the preparation of three-way automotive catalysts because of the ability of CeO₂ to store oxygen in form of lattice oxygen during catalyst oxidation stage and to release it during catalyst reduction step.

The prepared catalysts were tested in propane dehydrogenation reaction under different reaction conditions and thoroughly characterized by various complementary techniques even under in situ conditions. Density functional theory calculations were also carried out to derive insights into the mechanism of lattice oxygen diffusion from CeO₂ to V₂O₅ and into elementary pathways of propane dehydrogenation. The best-performing catalyst showed propene selectivity of above 90% at propane conversion of about 50% at 600°C and was stable over 300 dehydrogenation/reoxidation cycles. Such high performance is impressive. Although this study is well carried out, innovative and certainly has the potential to be attractive to a broad readership, the quality of data presentation/interpretation needs improvements and validation.

Critical comments

- i) The authors claim that propene formation occurs through the oxidative propane dehydrogenation with participation of catalyst lattice oxygen. I agree only partially with this statement due to following concerns. Using the results shown in Fig. 2b and assuming that the catalyst contains 10wt%V and 30wt%Ce (unfortunately, the caption of this figure does not contain this important information), I calculated the amount of lattice oxygen available for the oxidative dehydrogenation (V⁵⁺ is reduced to V³⁺ and Ce⁴⁺ is reduced to Ce³⁺) and the amount of propene formed during 65 min on propane stream. The latter amount is too high for the formation of propene through the oxidative propane dehydrogenation ($C_3H_8 + O = C_3H_6 + H_2O$) exclusively. Thus, propane must also be dehydrogenated non-oxidatively. My assumption is actually supported by the fact that H₂ was also detected (Fig.3g, Extended Data Figure 3). The authors should provide the amounts of H₂O and H₂ formed in their dehydrogenation tests (Fig.2b,d; Fig.3g; Extended Data Figure 1) and to estimate the contributions of oxidative and non-oxidative reactions.
- ii) Relatively high selectivity to CH₄, C₂H₄ and C₂H₆ (Extended Data Figure 1d) is typical for the non-oxidative propane dehydrogenation rather than for the oxidative dehydrogenation. If these products were formed through propane/propene oxidation, the discrepancy between the amounts of formed hydrocarbons and available catalyst lattice oxygen would be even large.
- iii) Considering comments (i) and (ii), the results of energy consumption calculations in Fig. 2e do not seem to be correct.

iv) The results shown in Extended Data Figure 1e indirectly support my concern (i). The oxidative dehydrogenation of propane is not thermodynamically limited. The rate of propene formation in this reaction is typically directly related to propane partial pressure, i.e., a first-order reaction with respect to propane. This means that propane conversion does not depend on propane partial pressure. Contrarily, the authors determined a decrease in the conversion. How can this dependence be explained? I assume that it is due to a strong contribution of non-oxidative propane dehydrogenation to propene formation.

v) How was the rate of propene formation in Extended Data Figure 1f determined? Under differential or integral reactor operation?

vi) I am not expert in DFT calculations but cannot understand why the oxidation states of Ce and V the model for ML-VO₂ under oxidizing conditions are 3+ and 4+, respectively. These metals should be in their highest oxidation state in air and lower their oxidation state in propane feed.

vii) How is water formed during propane dehydrogenation to propene? Why did the authors not calculate this pathway? According to Fig. 4d, ML-VO₂ should be more active than V₂O₅ (lower overall barrier). However, the initial (extrapolate to zero time on stream) propane conversion over VO_x is higher than over VO_x-CeO₂ (Fig.2b)

Specific minor comments

i) The authors calculated propene selectivity with and without consideration of coke formation. It is unclear from the manuscript which value is reported. Which selectivity was used to calculate propene yield? I assume that formulae 1-2a and 1-2b are mixed up.

ii) Time on stream profiles of water formed in dehydrogenation tests should be presented.

iii) Page 20. The meaning of “surface change coefficient (kchem)” is unclear. Sometime the authors write surface exchange coefficient. What does this coefficient really mean? How is it important for propene formation?

iv) Page 20. Some abbreviations in equations 2-5, 2-6, 2-7 is not explained.

v) Why do the authors not show the fits of TGA experiments? How did the authors take into account coke formation for fitting the obtained TGA profiles? According to Extended Data Figure 4a,c,e,g, the overall catalyst weight increases with time on propane stream.

vi) Pages 33, 34. Something is wrong with Extended Data Table 1.

v) Captions of some figures are confusing and do not contain catalyst composition. Please, check the manuscript.

vi) How can NH₃-TPD tests be used to determine the concentration of Lewis acidic sites exclusively?

vii) According to a recent review from the same research group (DOI: 10.1039/d0cs00814a), there are several Pt-containing non-oxidative dehydrogenation catalysts which showed very high propene selectivity at high (above 50%) degrees of propane conversion but are not mentioned in Fig.2c. Some relevant reference should be reported.

Some general comments related to the Abstract and Conclusion.

The conclusion paragraph is too general and does not seem to be really representative for this study. What does this phrase “diffusion and reaction of lattice oxygen species was demonstrated and kinetically coordinated” mean? Why and how kinetically coordinated? The authors have written “further improvements are expected for their exact atomic and electronic structure.” Which

improvements are expected? How can they be achieved?

Similar sentences are also written in the abstract. I am also wondering why the authors have written in the abstract "Propane dehydrogenation (PDH) has the potential to produce industrially important platform chemicals." Why does this reaction have the potential? Non-oxidative propane dehydrogenation is the basis of several large-scale processes.

Reviewer #2 (Remarks to the Author):

This manuscript written by Jinlong Gong et al. proposes ceria-vanadia core-shell catalysts as promising redox catalysts for propane dehydrogenation. The authors state that charged oxygen ions (O_2^-) generated from ceria oxygen carrier promote catalytic performance of ceria-vanadia catalysts for this reaction. Reliable characterization and adequate analyses are given in the manuscript. After reviewing this paper, I recommend that this paper is possible to be published in Nature Communications subjected to minor revisions. Some parts needed to be clarified. My comments are listed below:

1. In abstract, the authors state that "Charged oxygen ions (O_2^-) generated from ceria oxygen carrier was boosted to diffuse and transfer to vanadia dehydrogenation sites via a concerted hopping pathway at interface, stabilizing surface vanadia with moderate oxygen coverage at pseudo steady state for selective oxidative dehydrogenation without overoxidation or cracking." Using "without overoxidation or cracking" is too strong, this means no overoxidation or cracking should be observed. However, coking was observed in a ceria-vanadia catalyst as presented in Fig.3c.

2. On page 3, "Electron energy loss spectra (EELS) mappings of V L_{2,3} and Ce M_{4,5} edges affirmed vanadia sites anchored on widespread ceria nanodomains (Fig. 1h-k, Supplementary Fig. 1e-h)." There is no Fig. 1k in the main text.

3. On page 10, the authors explain that "When surface oxygen was fully removed and ML-VO₂ formed, oxygen diffusion from CeO₂ to VO_x turned to be endothermic with reaction energy of 0.62 eV. However, it became exothermic with further oxygen release in VO_x between ML-VO₂ and ML-V₂O₃, verifying continuous oxygen transfer from O(Ce) to O(V) sites (Supplementary Fig. 13)." How Supplementary Fig. 13 could explain "exothermic" and verify "continuous oxygen transfer from O(Ce) to O(V) sites"?

4. There are few comments for the DFT part.

- Propylene seems to be strongly attached on the catalyst's surface. How much are the desorption energies required for releasing propylene from the ceria-vanadia and V₂O₅ catalysts?

- The mechanistic studies of OPDH in V₂O₅ were reported. The authors should compare the results with literature. In addition, the authors investigated the stepwise mechanism. The authors should consider the concerted mechanism as well, especially in the ML-VO₂ model.

- The electronic charge analyses such as Bader charge or density of state (DOS) should be conducted to understand the valence states of atoms and electronic nature of studied catalysts.

- What is a criterion for the electronic convergence used in this work?

5. For the Fig.3c's caption, (top) and (bottom) should be specified in "(c) In situ Raman spectra of ceria-vanadia and vanadia at 600 °C".

There are typos such as "transient pules" in Fig. 3's caption and the main text.

In Fig. 4, each pathway should be labeled likes Extended Data Figure 8 or described in the caption.

Reviewer #3 (Remarks to the Author):

Nature Comm Review

Chen and co-workers have carried out a comprehensive investigation of vanadia on ceria nanoparticles for the selective oxidative dehydrogenation of propane. This report is a multi-scale investigation as the authors address atomic and molecular aspects of the catalyst performance—such as oxygen transport from ceria to vanadia—to process-wide analysis and comparison to existing propane dehydrogenation processes. However, there have been multiple investigations of vanadia/ceria catalysts (by well-known groups such as Freund in 2010 (10.1021/ja910574h) to more recent reports that investigate the structure and function of vanadia/ceria in-situ and at the molecular level (see this report by X. Wu et al. (10.1021/jacs.5b07939), by Hess et al (10.1021/jacs.2c06303), Iglesias-Juez (10.1021/acs.jpcc.7b09832), Taylor et al (10.1007/s11244-009-9307-0), and Sauer (10.1021/jp108185y)).

Chen et al have found how to prepare composite vanadia on ceria particles that seem to have the right dimensions and composition to work effectively in the context of a propane dehydrogenation cyclic process, which is a significant technical achievement, but not really a scientific one. In my opinion, the latest implementation of the CATOFIN propane dehydrogenation process is, functionally, similar to the process described in Figure 1. This latest process uses a dehydrogenation catalyst (chromia on alumina) AND a so-called Heat Generating Material (which I don't know what it is, but functions like the ceria in this report, by consuming hydrogen and producing heat, (<https://www.clariant.com/en/Business-Units/Catalysts/Petrochemical-and-Refining-Catalysts/CATOFIN-Technology>)). Also, as described in Figure 1, the proposed process is not formally a 'looping' process because not the catalyst or the reactants 'loop' around the process. This is a typical case of process intensification run in a cyclic process.

The paper is generally well written and logical in its presentation of information. The composite figures that are used throughout are not easy to follow and it would benefit from extending the description of the figures within the text (see Figure 3, for example). The text omits, in many instances, definite and indefinite articles and the readability of the paper would improve if this element were reviewed and corrected across the whole document.

For the reasons described above, I do not recommend this manuscript for publication in Nature Communications. A journal such as Applied Catalysis A or B, are a suitable outlet for this report, and will directly address the audience interested in this important industrial reaction.

Summary of revisions

Manuscript number: NCOMMS-22-53204-T

Questions	Revisions
Reviewer #1	
Comment 1: The authors should provide the amounts of H ₂ O and H ₂ formed in their dehydrogenation tests and to estimate the contributions of oxidative and non-oxidative reactions.	We have supplemented the sample detailed information in the captions, the formation of H ₂ O and H ₂ during the propane dehydrogenation test, and clarified the contribution of oxidative and non-oxidative dehydrogenation with the reaction time during the dehydrogenation step.
Comment 2: The results of energy consumption calculations in Fig. 2e do not seem to be correct.	For the energy consumption, we have considered the co-existence of H ₂ and H ₂ O, and the ratios of them were considered to be 70.2% and 29.8%, respectively, based on the H ₂ and H ₂ O distribution during the dehydrogenation step.
Comment 3: The authors determined a decrease in the conversion. How can this dependence be explained?	We have clarified the dthe supplemented the initial conversion and formation rate of propylene at 5 th min under differential reactor operation.
Comment 4: The authors should provide the amounts of H ₂ O and H ₂ formed in their dehydrogenation tests (Fig.2b, d; Fig.3g; Extended Data Figure 1) and to estimate the contributions of oxidative and non-oxidative reactions.	We have supplemented the sample detailed information in the captions, the formation of H ₂ O and H ₂ during the propane dehydrogenation test, and clarified the contribution of oxidative and non-oxidative dehydrogenation with the reaction time during the dehydrogenation step.
Comment 5: Contrarily, the authors determined a decrease in the conversion. How can this dependence be explained? How was the rate of propene formation in Extended Data Figure 1f determined? Under differential or integral reactor operation.	We have supplemented the initial propene formation rate at 5 th min to make the oxidative dehydrogenation more dominated under differential operation. C ₃ H ₆ formation rate showed a linear relationship with C ₃ H ₈ pressure, while C ₃ H ₈ conversion kept unity at different C ₃ H ₈ pressures, indicating that the rate of propene formation in this reaction is typically directly related to propane partial pressure.
Comment 6: How is water formed during propane dehydrogenation to propene? Why did the authors not calculate this pathway? According to Fig. 4d, ML-VO ₂ should be more active than V ₂ O ₅ (lower overall barrier).	We have supplemented the formation pathways of H ₂ O(g) and H ₂ (g) and compared them in the models of ML-VO ₂ and ML-V ₂ O ₃ . The conversion of ML-VO ₂ and V ₂ O ₅ (with high V loadings) was further compared.

Comment 7: It is unclear from the manuscript which selectivity is reported. Which selectivity was used to calculate propene yield? I assume that formulae 1-2a and 1-2b are mixed up.	We have corrected the propylene gas selectivity and total selectivity (including coke formation) in formulae 1-2a and 1-2b and clarified the reported propylene selectivity in the manuscript. The propylene yield was calculated based on propane conversion and propylene selectivity (including coke formation).
Comment 8: Time on stream profiles of water formed in dehydrogenation tests should be presented.	We thank the reviewer for this positive evaluation. We have supplemented the formation of H₂O and ratios of H₂/H₂O as a function of reaction time.
Comment 9: The meaning of “surface exchange coefficient (k_{chem})” is unclear. Sometime the authors write surface exchange coefficient. What does this coefficient really mean? How is it important for propene formation?	We have unified the writing of surface exchange coefficient (k_{chem}) and clarified its importance to propane activation and propylene formation.
Comment 10: Why do the authors not show the fits of TGA experiments? How did the authors take into account coke formation for fitting the obtained TGA profiles? According to Extended Data Figure 4a,c,e,g, the overall catalyst weight increases with time on propane stream?	We have supplemented the fitted curves in the TG profiles. To weaken the influence of coke formation and estimate the kinetics of oxygen release, we fitted the data by considering the stage of weight loss on ceria (30CeAl) and ceria-vanadia (6V/30CeAl) at the temperature range from 575 °C to 600 °C.
Comment 11: Some relevant Pt-containing non-oxidative dehydrogenation catalysts reference should be reported in Fig.2c.	Pt-containing non-oxidative dehydrogenation catalysts have been included in the updated results in Fig. 2c and Extended Data Table 2.
Comment 12: The conclusion and abstract paragraph is too general and does not seem to be really representative for this study.	We have carefully revised the abstraction, conclusion and improved the article accordingly.

Reviewer #2	
Comment 1: Using “without overoxidation or cracking” is too strong, this means no overoxidation or cracking should be observed. However, coking was observed in a ceria-vanadia catalyst as presented in Fig.3c.	We have revised the statement “without overoxidation or cracking” to “without significant overoxidation or cracking”.
Comment 2: How Supplementary Fig. 13 could explain “exothermic” and verify “continuous oxygen transfer from O(Ce) to O(V) sites”?	We have explained the term of “exothermic” and verified “continuous oxygen transfer from O(Ce) to O(V) sites”
Comment 3: Propylene seems to be strongly attached on the catalyst’s surface. How much are the desorption energies required for releasing propylene from the ceria-vanadia and V₂O₅ catalysts?	We have calculated the adsorption energies of propylene on ceria-vanadia and V₂O₅ catalysts, which showed the comparable values.
Comment 4: The mechanistic studies of OPDH in V₂O₅ were reported. The authors should compare the results with literature. In addition, the authors investigated the stepwise mechanism. The authors should consider the concerted mechanism as well, especially in the ML-VO₂ model.	We have compared our results with the literature (ACS Catal. 2019, 9, 5816–5827), which are very similar. For the concerted mechanism, we want to highlight the proposed concerted oxygen transfer mechanism is calculated in ML-VO₂ model indeed.
Comment 5: The electronic charge analyses such as Bader charge or density of state (DOS) should be conducted to understand the valence states of atoms and electronic nature of studied catalysts.	We have supplemented the Bader charge analysis to further prove that CeO₂ was reduced to preserve the valence states of vanadia before ML-V₂O₃ state.
Comment 6: What is a criterion for the electronic convergence used in this work?	We have provided the criterion for the electronic convergence that used in this work and the value is 10⁻⁴ eV.

Comment 7: For the Fig.3c's caption, (top) and (bottom) should be specified in "(c) In situ Raman spectra of ceria-vanadia and vanadia at 600 °C". There are typos such as "transient pulses" in Fig. 3's caption and the main text. In Fig. 4, each pathway should be labeled likes Extended Data Figure 8 or described in the caption.	We have revised the captions in Fig 3. and labeled each reaction pathway in Fig. 4.
Reviewer #3	
Comment 1: There have been multiple investigations of vanadia/ceria catalysts that investigate the structure and function of vanadia/ceria in-situ and at the molecular level.	The research status and remaining open questions of vanadia/ceria catalysts have been supplemented in the introduction part.
Comment 2: The latest implementation of the CATOFIN propane dehydrogenation process is, functionally, similar to the process described in Figure 1.	We have clarified the two main differences of CATOFIN process loaded with HGM and showed the ceria-vanadia core-shell redox catalysts exhibited higher propylene selectivity and lower deactivation rate compared with the physical mixture of dehydrogenation catalyst VO_x with solid oxygen carrier CeO₂,
Comment 3: As described in Figure 1, the proposed process is not formally a 'looping' process because not the catalyst or the reactants 'loop' around the process. This is a typical case of process intensification run in a cyclic process?	Ceria-vanadia redox catalysts serve as the oxygen carriers by donating lattice oxygen from oxidation states (V⁵⁺, V⁴⁺, Ce⁴⁺) to propane in the dehydrogenation (reduction) step in the redcuer, affording a reduced valence state (V³⁺, Ce³⁺) that can be reoxidized in the regeneration (oxidation) step by air in the oxidizer to close the loop as showed in Fig. 1a.
Comment 4: It would benefit from extending the description of the figures within the text (see Figure 3, for example). The text omits, in many instances, definite and indefinite articles and the readability of the paper would improve if this element were reviewed and corrected across the whole document.	We have fully extended the descriptions of the figures within the text. Additionally, the articles and relevant sentences have been carefully and fully revised throughout the manuscript.

Reviewer #1:

General comments R1: *Conversion of available but chemically inert short alkanes into value-added chemicals is an important topic both from fundamental and application viewpoints. The present manuscript reports about propane dehydrogenation to propene over V-Ce-containing catalysts under O₂-free conditions. Such operation mode is called as chemical looping approach, according to which the reaction proceeds oxidatively with participation of catalyst lattice oxygen. The group of Jinlong Gong is well known for the activity in this area. In this study, the authors prepared core(CeO₂)-shell(V₂O₅) catalysts based on Al₂O₃ support. The idea is to tune activity and selectivity of V₂O₅ through the redox properties of ceria. A similar idea is used for the preparation of three-way automotive catalysts because of the ability of CeO₂ to store oxygen in form of lattice oxygen during catalyst oxidation stage and to release it during catalyst reduction step. The prepared catalysts were tested in propane dehydrogenation reaction under different reaction conditions and thoroughly characterized by various complementary techniques even under in situ conditions. Density functional theory calculations were also carried out to derive insights into the mechanism of lattice oxygen diffusion from CeO₂ to V₂O₅ and into elementary pathways of propane dehydrogenation. The best-performing catalyst showed propene selectivity of above 90% at propane conversion of about 50% at 600°C and was stable over 300 dehydrogenation/reoxidation cycles. Such high performance is impressive. Although this study is well carried out, innovative and certainly has the potential to be attractive to a broad readership, the quality of data presentation/interpretation needs improvements and validation.*

Response: We thank the reviewer for this positive evaluation. We have carefully addressed the comments raised by the reviewer and revised the manuscript accordingly.

Specific Comments R1-1: *The authors claim that propene formation occurs through the oxidative propane dehydrogenation with participation of catalyst lattice oxygen. I agree only partially with this statement due to following concerns. Using the results shown in Fig. 2b and assuming that the catalyst contains 10wt%V and 30wt%Ce (unfortunately, the caption of this figure does not contain this important information), I calculated the amount of lattice oxygen available for the oxidative dehydrogenation (V⁵⁺ is reduced to V³⁺ and Ce⁴⁺ is reduced to Ce³⁺) and the amount of propene formed during 65 min on propane stream. The latter amount is too high for the formation of propene through the oxidative propane dehydrogenation (C₃H₈ + O = C₃H₆ + H₂O) exclusively. Thus, propane must also be dehydrogenated non-oxidatively. My assumption is actually supported by the fact that H₂ was also detected (Fig.3g, Extended Data Figure 3). The authors should provide the amounts of H₂O and H₂ formed in their dehydrogenation tests to estimate the contributions of oxidative and non-oxidative reactions.*

Response: We thank the reviewer for the useful and constructive comments and appreciate the suggestions that both of oxidative and non-oxidative dehydrogenation made contribution to propylene formation. We would have claimed that the production of propylene mainly through oxidative dehydrogenation without significant overoxidation, or cracking in the non-oxidative dehydrogenation period. We have carefully revised the relevant statements in the manuscript.

Additionally, we have supplemented the sample detailed information in the captions, the formation of H₂O and H₂, and clarified the contribution of oxidative and non-oxidative dehydrogenation during the dehydrogenation step.

As the reviewer suggested that due to the release of lattice oxygen in ceria-vanadia redox catalysts with the reaction time, V and Ce would be reduced to be V³⁺ and Ce³⁺, and propane dehydrogenation transformed from oxidative dehydrogenation to non-oxidative dehydrogenation. We have calculated the amount of lattice oxygen available in ceria-vanadia redox catalysts (0.5 g loadings, 6V/30CeAl, 6 wt% V and 30 wt% Ce) (V⁵⁺ is reduced to V³⁺ and Ce⁴⁺ is reduced to Ce³⁺) to produce propene entirely through the oxidative propane dehydrogenation (C₃H₈ + O = C₃H₆ + H₂O) without undergoing non-oxidative dehydrogenation at the flow rate of 4 mL/min C₃H₈. The time was determined to be about 6.3 min if the available oxygen was fully consumed. Under our reaction test, the propane conversion was less than 50%, therefore longer time than 12.6 min could be achieved. Additionally, based on the previous results (J. Am. Chem. Soc. 2019, 141, 18653-18657; ACS Catal. 2016, 6, 5207-5214.), the reduced V³⁺ could functionalize as the catalytic active sites for non-oxidative dehydrogenation that contribute to propane conversion. Therefore, in addition to the oxidative dehydrogenation, non-oxidative dehydrogenation could also be existed as suggested by the reviewer.

According to the review's comments, we have supplemented the formation of H₂O and H₂ in their dehydrogenation tests and estimated the contributions of oxidative and non-oxidative reactions in Figure R1-1. As shown in the results, the initial ratio of H₂O to H₂ at 5th min was determined to be 0.44, however, it would decrease to be about 0.05 after 60 min. Therefore, oxidative dehydrogenation was dominated in the initial period and it would transform to the period of non-oxidative dehydrogenation with reaction time, especially after 60-min test. Therefore, the re-oxidation step was introduced to recover the lattice oxygen after 30-min dehydrogenation test. During the continuous dehydrogenation-reoxidation cycles, the ratios of H₂O to H₂ maintained at about 0.21, as showed in Figure R1-1b.

Figure R1-1. (a) Propane conversion, propylene selectivity and ratios of H₂O/H₂ as a function of time over ceria-vanadia redox catalysts (6V/30CeAl). Conditions: 600 °C, GHSV =2,500 h⁻¹, C₃H₈/N₂=0.25. (b) Cyclic performance over ceria-vanadia redox catalysts (6V/30CeAl). (Dehydrogenation step: 600 °C, GHSV =2,500 h⁻¹, C₃H₈/N₂=0.25 for 30 minutes; Inert purge: 600

°C, N₂=40 mL/min for 5 minutes; Oxidation step: 600 °C, 20 vol.% O₂/N₂=20 mL/min for 15 mins).

To catch the transient distributions of products, C₃H₆, CO_x, H₂O and H₂, especially in the initial period (<5 min), C₃H₈-pulse experiments using on-line MS were employed (Extended Data Figure 4). CO_x was firstly observed due to the overoxidation on the active surface oxygen species (period I), then it would undergo selectively oxidative dehydrogenation to C₃H₆ and H₂O caused by lattice oxygen (period II). After the lattice oxygen were fully released, cracking or hydrogenolysis occurred which was more dominated for pure vanadia catalysts, leading to the formation of CH₄ (period III). F_{H₂/H₂O} (the intensity ratios of H₂ and H₂O) was defined to semi-quantify the contribution of oxidative and non-oxidative dehydrogenation during the dehydrogenation step. The results clearly evidenced the co-formation of C₃H₆ and H₂O through the oxidative propane dehydrogenation (C₃H₈ + O = C₃H₆ + H₂O) in the initial pulses and then the formation of H₂ through the non-oxidative propane dehydrogenation (C₃H₈ = C₃H₆ + H₂) with pulse numbers, which was more dominated (period II and III) for sole vanadia catalysts due to the limited oxygen availability.

Extended Data Figure 4 | Surface reaction routes determined by the temperature-programmed surface reaction and transient pulse experiments. Product distributions at 600 °C and the calculated F_{H₂/H₂O}, which is the intensity ratio of H₂/H₂O obtained from the pulse profiles, (b, c) vanadia (6V/Al), (e, f) ceria-vanadia (6V/30CeAl) and (h, i) ceria (30CeAl) catalysts. C₃H₈, C₃H₆, CO₂,

CH₄, H₂, and H₂O, m/e equals 29, 41, 44, 16, 2 and 18, respectively.

Therefore, in the initial period with sufficient lattice oxygen, propane oxidative dehydrogenation or mixed oxidative and non-oxidative dehydrogenation were dominated, however when the lattice oxygen was fully reduced, non-oxidative dehydrogenation would be dominated. The diagram of redox catalysts in propane dehydrogenation by the chemical looping engineering in Figure 1a has been revised that C₃H₆, H₂O and H₂ are the main products at the outlet of the dehydrogenation reactor (reducer).

Figure 1. (a) The diagram of redox catalysts in propane dehydrogenation by chemical looping engineering.

In page 10

Evidenced by the in situ spectroscopies, the release of lattice oxygen would induce the existence and transformation of different reaction periods, including overoxidation, oxidative dehydrogenation, and non-oxidative dehydrogenation. Under differential reactor operation by controlling the propane conversion lower than 10%, C₃H₆ formation rate showed a linear relationship with C₃H₈ pressure while C₃H₈ conversion kept identical at different C₃H₈ pressures, indicating that the rate of propene formation in this reaction is typically directly related to propane partial pressure, i.e., a first-order reaction with respect to propane. To clarify the contribution of oxidative and non-oxidative dehydrogenation, the formation of H₂O and H₂ over ceria-vanadia redox catalysts in their dehydrogenation tests were investigated. As shown in Extended Fig. 3, the initial ratio of H₂O to H₂ at 5th min was determined to be 0.44, however, it would decrease to be about 0.05 after 60 min. Therefore, oxidative dehydrogenation was dominated in the initial period and it was transformed to non-oxidative dehydrogenation with reaction time, accounting for the introduction of re-oxidation step to recover the lattice oxygen after 30-min dehydrogenation test during the continuous dehydrogenation-reoxidation cycles with the ratios of H₂O to H₂ of ~ 0.21.

To catch the transient distributions of products, C₃H₆, CO_x, H₂O and H₂, especially in the initial period, C₃H₈ transient pulse experiments were employed at 600 °C using on-line mass spectrometry (MS) (Extended Data Figure 4). CO_x was firstly observed due to the overoxidation on the active surface oxygen species (period I), then it would undergo selectively oxidative dehydrogenation to C₃H₆ and H₂O caused by lattice oxygen (period II). After the lattice oxygen were fully released, cracking or hydrogenolysis occurred which was more

dominated for pure vanadia catalysts, leading to the formation of CH_4 (period III). Intensity ratios of $\text{H}_2/\text{H}_2\text{O}$ defined as $F_{\text{H}_2/\text{H}_2\text{O}}$, a sign of H combustion, maintained at five over ceria-vanadia redox catalysts (Fig. 3g), implying that O^{2-} diffusion from ceria oxygen carrier leads to a pseudo-steady-state H combustion at surface vanadia sites^{6,31}. In contrast, pure vanadia showed stage-divided products, along with the tenfold increased intensity ratios of $\text{H}_2/\text{H}_2\text{O}$ with pulses of C_3H_8 , indicating the presence of O^{2-} gradient and transport limitation in the pure vanadia³¹. Gradually decreased ratios of $\text{H}_2/\text{H}_2\text{O}$ in pure ceria implied surface H abstraction were sluggish due to the less active C–H dissociation center of CeO_2 . We then designed a temperature-programmed surface reaction (TPSR) to explore the surface reaction route. Initial dehydrogenation temperatures of C_3H_8 ($m/z=29$) on pure vanadia and ceria-vanadia redox catalysts were at 338 and 340 °C, respectively, lower than that of pure ceria (412 °C) (Extended Data Fig. 3), indicating the presence of low-temperature C–H dissociation V–O centers ascribed from active vanadia catalysts, corresponding to increased medium acidic sites as shown in NH_3 -TPD profiles (Supplementary Fig. 11).

In Page 28

Extended Data Figure 3 | The formation of H_2O and H_2 , and C_3H_6 formation rates. (a) Propane conversion, propylene selectivity and ratios of $\text{H}_2\text{O}/\text{H}_2$ as a function of time on stream over ceria-vanadia redox catalysts (6V/30CeAl). Conditions: 600 °C, GHSV = 2,500 h^{-1} , $\text{C}_3\text{H}_8/\text{N}_2=0.25$. (b) Cyclic performance over ceria-vanadia redox catalysts (6V/30CeAl). (Dehydrogenation step: 600 °C, GHSV = 2,500 h^{-1} , $\text{C}_3\text{H}_8/\text{N}_2=0.25$ for 30 minutes; Inert purge: 600 °C, $\text{N}_2=40$ mL/min for 5 minutes; Oxidation step: 600 °C, 20 vol.% $\text{O}_2/\text{N}_2=20$ mL/min for 15 minutes). (c) C_3H_6 formation rates and C_3H_8 conversion as a function of C_3H_8 pressure (diluted by N_2) at 600 °C. (d) C_3H_8 conversion and C_3H_6 selectivity as a function of time on 40V/Al, 6V/Al and 6V/30CeAl at 600 °C. Reaction test conditions: 600 °C, 1.4 atmospheric pressure, GHSV=2500 h^{-1} , 0.5 g of sample, $\text{C}_3\text{H}_8/\text{N}_2 = 0.25$.

Fig. 1 | Identification of vanadia layers that coat ceria nanodomains. (a) Diagram of core-shell redox catalysts in propane dehydrogenation by the chemical looping engineering: dehydrogenation and oxidation in fuel reactor (reducer) and air reactor (oxidizer), respectively. (b-d) HAADF-STEM images and (e-h) EELS mappings of core-shell ceria-vanadia redox catalysts (6V/30CeAl): (f): V; (g): Ce; (h): V + Ce. (i) Line-scanning EELS. (j) EELS of the domains ((1), (2), (3)) in (b).

Specific Comments R1-2: *Relatively high selectivity to CH₄, C₂H₄ and C₂H₆ (Extended Data Figure 1d) is typical for the non-oxidative propane dehydrogenation rather than for the oxidative dehydrogenation. If these products were formed through propane/propene oxidation, the discrepancy between the amounts of formed hydrocarbons and available catalyst lattice oxygen would be even large.*

Response: We are grateful to the reviewer for the valuable comments. As the reply to the first comment, we agree that both of oxidative and non-oxidative dehydrogenation made contribution to propylene formation and non-oxidative propane dehydrogenation would be more dominated when oxygen was fully released. We should mention that the data in Extended Data Figure 1d were obtained at 3-30 min, during which oxidative and non-oxidative dehydrogenation could be co-existed. To demonstrate the product evolutions with reaction time, especially in the very initial period less than 3 min, C₃H₈-pulses using on-line MS (Extended Data Figure 4b, e, h) was employed, which showed that, CO_x was firstly observed due to the overoxidation on the active surface oxygen species, then it would undergo selectively oxidative dehydrogenation to C₃H₆ and H₂O caused by lattice oxygen. After the lattice oxygen were fully released, cracking or hydrogenolysis occurred which was more dominated for pure vanadia catalysts, leading to the formation of CH₄. And the CH₄ selectivity on pure vanadia catalysts were determined to be about 25% at 5th min, which is much higher than that on ceria-vanadia redox catalysts (~5%), indicating the efficient usage of lattice oxygen caused by vanadia-ceria interaction without the significant cracking in the non-oxidative dehydrogenation period. Therefore, the formation of CH₄, C₂H₄ and C₂H₆ would be more dominated if the cracking or hydrogenolysis was dominated with oxygen release.

Specific Comments R1-3: *Considering comments (i) and (ii), the results of energy consumption calculations in Fig. 2e do not seem to be correct.*

Response: We thank the reviewer for this comment. For the energy consumption, we have considered the co-existence of H₂ and H₂O. Average 70.2% H₂ and 29.8% H₂O were considered to calculate the energy consumption (Table S7) based on the H₂ and H₂O distributions during the dehydrogenation step.

Table S7. Simulation Settings (feeds and yields).

	Oleflex	Chemical Looping PDH
Feed wt%	37500 kg/hr (96 propane, 2 ethane, 2 butane)	
Carbon yield mole% (propane-based)		
Propane	65.00	50.00
Propylene	31.50	43.30
Methane	0.93	2.25
Ethylene	0.93	1.13
Ethane	0.94	1.28

Allylene	0.70	0
Carbon dioxide	0	0.57
Coke(graphite)	0	1.50
Hydrogen yield mole% (propane-based)		
Hydrogen	76.96	70.20
Water	23.04	29.80

Specific Comments R1-4: *The results shown in Extended Data Figure 1e indirectly support my concern (i). The oxidative dehydrogenation of propane is not thermodynamically limited. The rate of propene formation in this reaction is typically directly related to propane partial pressure, i.e., a first-order reaction with respect to propane. This means that propane conversion does not depend on propane partial pressure. Contrarily, the authors determined a decrease in the conversion. How can this dependence be explained? I assume that it is due to a strong contribution of non-oxidative propane dehydrogenation to propene formation?*

Response: We thank the reviewer for the comments. As the reply to the first comment, we agree that both of oxidative and non-oxidative dehydrogenation made contribution to propylene formation and non-oxidative propane dehydrogenation would be more dominated when oxygen was fully released. The conversion of propane and propene formation rates were determined under the integral reactor operation at high propane conversion (33.7%-57.8%) as showed in Extended Data Figure 1e. Therefore, propane conversion would decrease with higher WHSV of propane.

To investigate the dependence of propylene formation rates and propane conversion on propane pressure under differential reactor operation, we have supplemented the initial propene formation rate at 5th min to make the oxidative dehydrogenation more dominated under the differential operation by controlling the propane conversion lower than 10%, which was shown in Figure R1-2. C₃H₆ formation rate showed a linear relationship with C₃H₈ pressure, while C₃H₈ conversion kept identical at different C₃H₈ pressures, indicating that the rate of propene formation in this reaction is typically directly related to propane partial pressure, i.e., a first-order reaction with respect to propane. The updated results are shown in Extended Data Figure 3c.

Figure R1-2. C₃H₆ formation rates and C₃H₈ conversion as a function of C₃H₈ pressure (diluted by N₂) at 600 °C.

Extended Data Figure 3 | The formation of H₂O and H₂, and C₃H₆ formation rates. (a) Propane conversion, propylene selectivity and ratios of H₂O/H₂ as a function of time on stream over ceria-vanadia redox catalysts (6V/30CeAl). Conditions: 600 °C, GHSV = 2,500 h⁻¹, C₃H₈/N₂ = 0.25. (b) Cyclic performance over ceria-vanadia redox catalysts (6V/30CeAl). (Dehydrogenation step: 600 °C, GHSV = 2,500 h⁻¹, C₃H₈/N₂ = 0.25 for 30 minutes; Inert purge: 600 °C, N₂ = 40 mL/min for 5 minutes; Oxidation step: 600 °C, 20 vol.% O₂/N₂ = 20 mL/min for 15 minutes). (c) C₃H₆ formation rates and C₃H₈ conversion as a function of C₃H₈ pressure (diluted by N₂) at 600 °C. (d) C₃H₈ conversion and C₃H₆ selectivity as a function of time on 40V/Al, 6V/Al and 6V/30CeAl at 600 °C. Reaction test conditions: 600 °C, 1.4 atmospheric pressure, GHSV = 2500 h⁻¹, 0.5 g of sample, C₃H₈/N₂ = 0.25.

Specific Comments R1-5: How was the rate of propene formation in Extended Data Figure 1f determined? Under differential or integral reactor operation?

Response: We thank the reviewer for the valuable comment. We have supplemented the clarification in the caption. The rate of propene formation in Extended Data Figure 1f was determined under the integral reactor conditions, which was the formation rate of propene at the reactor exit. Therefore, the propane conversion would decrease with higher WHSV of propane. However, under the differential operations, C₃H₆ formation rate showed a linear relationship with C₃H₈ pressure, while C₃H₈ conversion kept identical at different C₃H₈ pressures.

Extended Data Figure 1 | Schematic cyclogram, and effects of V contents, Ce contents, WHSV of propane, and temperature, on propane conversion, propylene selectivity, and formation rates. Effect of WHSV of propane on (e) propane conversion, propylene selectivity, and (f) propylene formation rates over ceria-vanadia redox catalysts (6V/30CeAl) at the reactor exit. Reaction test conditions: WHSV of propane = 0.5, 1, 2, 4 h⁻¹ (C₃H₈: 10%, 20%, 40%, 80% vol diluted by N₂), 600 °C, and 1.4 atmospheric pressure. ...

Specific Comments R1-6: I am not expert in DFT calculations but cannot understand why the oxidation states of Ce and V the model for ML-VO₂ under oxidizing conditions are 3+ and 4+, respectively. These metals should be in their highest oxidation state in air and lower their oxidation state in propane feed.

Response: We thank the reviewer for the comments. In the DFT calculation part, we aim at simulating the reduction process of our catalyst in propane feed by establishing a series of model with different reduction degrees (ML-V₂O₅, ML-VO₂, and ML-V₂O₃). For ML-V₂O₅ in air, the valence states of V and Ce were highest oxidation states of +5 and +4, respectively. ML-VO₂ is the intermediate model and the vanadia in this model is already reduced in propane feed. Therefore, the oxidation states should be lower than their highest oxidation states.

Specific Comments R1-7: How is water formed during propane dehydrogenation to propene? Why did the authors not calculate this pathway? According to Fig. 4d, ML-VO₂ should be more active than V₂O₅ (lower overall barrier). However, the initial (extrapolate to zero time on stream) propane conversion over VO_x is higher than over VO_x-CeO₂ (Fig.2b).

Response: We thank the reviewer for the comment. We have supplemented the formation pathways of H₂O(g) and H₂(g) and compared them in the models of ML-VO₂ and ML-V₂O₃. As shown in Figure R1-3a, in the model of ML-VO₂, the main adsorption site of H* is V=O site. The activation barrier of formation H₂* intermediate is low while the formed H₂* has a stronger preference to form H₂O(g) rather than H₂(g). Such results denote ML-VO₂ is still in the dominant period of oxidative dehydrogenation. When the vanadia is further reduced, ML-V₂O₃ was existed. In Figure R1-3b, we could observe that all V=O sites are removed, and main adsorption site of H* becomes V-O-V. The formation activation barrier of H₂* is higher compared with the barrier in ML-VO₂, indicating the elimination of H* and regeneration of active site are more difficult. Further, H₂* tends to form H₂(g) at this stage, denoting the model already enters the dominant period of non-oxidative dehydrogenation. According to our experimental results that in the initial period with sufficient lattice oxygen, propane oxidative dehydrogenation was dominated. However, when the lattice oxygen was fully reduced, non-oxidative dehydrogenation would be dominated.

Figure R1-3. Reaction energy profiles of H₂O formation and H₂ formation over (a) ML-VO₂ and (b) ML-V₂O₃.

For the activation barrier of first dehydrogenation step of propane that concerns the conversion, 6V/Al (6 wt% V, the same loading of V with ceria-vanadia catalysts (6V/30CeAl)) was used before as the reference sample to compare the conversion instead of the crystalline V_2O_5 model that used in the DFT calculations. Previous results (10.1021/acscatal.6b00893; 10.1002/cctc.201500151) have showed that lower V loadings would lead to the better dispersion of VO_x on the surface of Al_2O_3 which showed higher propane dehydrogenation reactivity and selectivity than crystalline V_2O_5 . We have prepared the VO_x/Al_2O_3 with high V loading of 40 wt%, and the formation of crystalline V_2O_5 was evidenced by XRD compared with the V_2O_5 standard (PDF#41-1426) (Figure R1-4a). As the results showed in Figure R1-4b, 40V/Al (crystalline V_2O_5) showed lower initial propane conversion at 5th min and propylene selectivity than 6V/30CeAl (ML- VO_2), which is corresponding with the DFT calculations that ML- VO_2 showed lower overall barrier than crystalline V_2O_5 .

Figure R1-4. (a) XRD patterns of 40V/Al, 6V/Al and 6V/30CeAl. (b) C_3H_8 conversion and C_3H_6 selectivity as a function of time on 40V/Al, 6V/Al and 6V/30CeAl at 600 °C. Reaction test conditions: 600 °C, 1.4 atmospheric pressure, GHSV=2500 h^{-1} , 0.5 g of sample, $C_3H_8/N_2 = 0.25$.

In page 13

Valence states of V have been believed to manipulate propylene selectivity^{35,36}. The following calculations on the formation pathways of $H_2O(g)$ and $H_2(g)$ showed that ML- VO_2 has a strong preference to proceed oxidative dehydrogenation compared with ML- V_2O_3 (Supplementary Fig. 16) and exhibited lower barriers of first and second dehydrogenation of propane to propylene, but higher barriers for acetone formation, a significant intermediate of overoxidation, than crystalline V_2O_5 and ML- V_2O_3 (Fig. 4d, Extended Data Fig. 9), according to experimental results that ceria-vanadia redox catalysts showed higher initial propane conversion and propylene selectivity than 6V/Al and crystalline V_2O_5 . These results firmly support that lattice oxygen transfer from ceria was to stabilize moderate V valence states and oxygen coverage for selective dehydrogenation without significant overoxidation or cracking and coking.

Figure S16. / Reaction energy profiles of H_2O formation and H_2 formation over (a) $\text{ML-V}\text{O}_2$ and (b) $\text{ML-V}_2\text{O}_3$.

The result indicates in $\text{ML-V}\text{O}_2$ period, the main adsorption sites for H is $\text{V}=\text{O}$ site. The activation barrier of formation H_2^* intermediate is low and the formed H_2^* has a strong preference to form $\text{H}_2\text{O}(\text{g})$ rather than $\text{H}_2(\text{g})$. Such results denote $\text{ML-V}\text{O}_2$ is still in the CL-ODH dominant period. When the vanadia is further reduced, the model enters $\text{ML-V}_2\text{O}_3$ period. In Figure S13 (b), we could observe all $\text{V}=\text{O}$ is removed, and main adsorption site becomes $\text{V}-\text{O}-\text{V}$. The formation activation barrier of H_2^* is higher compared with the barrier in $\text{ML-V}\text{O}_2$, indicating the elimination of H^* and regeneration of active site are more difficult. Further, H_2^* tends to form $\text{H}_2(\text{g})$ in this period, denoting the model already enters PDH dominant period.

In Page 28

Extended Data Figure 3 / The formation of H_2O and H_2 , and C_3H_6 formation rates. (a) Propane conversion, propylene selectivity and ratios of $\text{H}_2\text{O}/\text{H}_2$ as a function of time on stream over ceria-vanadia redox catalysts (6V/30CeAl). Conditions: 600°C , $\text{GHSV} = 2,500 \text{ h}^{-1}$, $\text{C}_3\text{H}_8/\text{N}_2 = 0.25$. (b) Cyclic performance over ceria-vanadia redox catalysts (6V/30CeAl). (Dehydrogenation step: 600°C , $\text{GHSV} = 2,500 \text{ h}^{-1}$, $\text{C}_3\text{H}_8/\text{N}_2 = 0.25$ for 30 minutes; Inert purge: 600°C , $\text{N}_2 = 40 \text{ mL/min}$ for 5 minutes; Oxidation step: 600°C , 20 vol.% $\text{O}_2/\text{N}_2 = 20 \text{ mL/min}$ for 15 minutes). (c) C_3H_6 formation rates and C_3H_8 conversion as a function of C_3H_8 pressure (diluted by N_2) at 600°C . (d) C_3H_8 conversion and C_3H_6 selectivity as a function of time on 40V/Al, 6V/Al and 6V/30CeAl at 600°C . Reaction test conditions: 600°C , 1.4 atmospheric pressure, $\text{GHSV} = 2500 \text{ h}^{-1}$, 0.5 g of sample, $\text{C}_3\text{H}_8/\text{N}_2 = 0.25$.

Specific Comments R1-8: *The authors calculated propene selectivity with and without consideration of coke formation. It is unclear from the manuscript which value is reported. Which selectivity was used to calculate propene yield? I assume that formulae 1-2a and 1-2b are mixed up.*

Response: We thank the reviewer for the valuable comments. We have corrected the propylene gas selectivity and total selectivity (including coke formation) in formulae 1-2a and 1-2b and clarified the reported propylene selectivity in the manuscript. The propylene selectivity (including coke formation) was shown in Fig. S4 and the C balance was shown in Extended Figure 1d. The propylene yield was calculated based on propane conversion and propylene selectivity (including coke formation). For ceria-vanadia redox catalysts (6V/30CeAl), due to the effective usage of lattice oxygen, the propylene gas selectivity and total selectivity (including coke formation) are 88.6% and 85.1% respectively at 5th min and close to 89% at propane conversion of 49% at 30th min during the dehydrogenation step. In contrast, sole vanadia catalysts (6V/Al) showed 71.6% propylene gas selectivity and 57.2% propylene selectivity (including coke formation) at 5th min due to the significant coke formation.

Specific Comments R1-9: *Time on stream profiles of water formed in dehydrogenation tests should be presented.*

Response: We thank the reviewer for this positive evaluation. We have supplemented the formation of H₂O and ratios of H₂/H₂O as a function of reaction time, as showed in Figure R1-1a.

Figure R1-1a. Propane conversion, propylene selectivity and ratios of H₂O/H₂ as a function of time on stream on ceria-vanadia redox catalysts (6V/30CeAl). Conditions: 600 °C, GHSV =2,500 h⁻¹, C₃H₈/N₂=0.25.

Specific Comments R1-10: *Page 20. The meaning of “surface change coefficient (k_{chem})” is unclear. Sometime the authors write surface exchange coefficient. What does this coefficient really mean? How is it important for propene formation?*

Response: We thank the reviewer for this comment. We have unified the writings of surface exchange coefficient (*k_{chem}*). Based on previous reports (10.1039/c4cp05742b (2015);

10.1016/j.cej.2021.130417 (2021)), we assume, in a first approximation, that it is sufficient to model the bulk diffusion and surface reaction process in the oxygen carrier particles by two constant parameters bulk diffusion coefficient (D_{diff}) and surface exchange coefficient (k_{chem}), where D_{diff} is an effective chemical oxygen diffusion coefficient and k_{chem} is an effective chemical surface exchange coefficient of oxygen involving the first order surface reaction at the gas/solid interface (J. Crank, *The Mathematics of Diffusion*, Oxford University Press, 1975.). The advantage of such a simple model is that it allows one to derive analytical solutions which yield significant insight into the role the microstructure of a given material has on the overall kinetics of the bulk diffusion–surface reaction process.

In turn, k_{chem} can reflect the surface oxygen reactivity at the gas/solid interface. Higher k_{chem} would lead to lower C-H activation energy barriers and higher forward rates to produce propene. Compared to ceria, ceria-vanadia catalysts showed about 10 times higher k_{chem} and lower surface exchange barrier (89.8 vs. 176.3 kJ/mol), implying the higher surface dehydrogenation reaction rates on ceria-vanadia catalysts.

Specific Comments R1-11: *Page 20. Some abbreviations in equations 2-5, 2-6, 2-7 is not explained?*

Response: We thank the reviewer for this reminder. The abbreviations in equations 2-5, 2-6, 2-7 have been fully explained.

$$\frac{C_o(t,r)-C_o(\infty)}{C_o(0)-C_o(\infty)} = \frac{2LR}{r} \sum_{n=1}^{\infty} \frac{e^{\left(-D_{diff} \frac{\beta_n^2 t}{R^2}\right)}}{[\beta_n^2 + L^2 - L]} \cdot \frac{\sin\left(\beta_n \frac{r}{R}\right)}{\sin(\beta_n)} \quad (2-5)$$

with

$$\beta_n \cot(\beta_n) + L - 1 = 0 \quad (2-6)$$

$$L = \frac{Rk_{chem}}{D_{diff}} \quad (2-7)$$

where $C_o(\infty)$ is the oxygen species concentration at infinite reaction time; $C_o(t,r)$ is the oxygen species concentration at reaction time t ; C_o is the oxygen species concentration in the catalysts; t is the reaction time; r is the radius at the specific position of the catalysts powder; R is the average radius of the catalyst powder. β_n is dimensionless parameter. L is characteristic length of the solid one; k_{chem} is surface exchange coefficient; D_{diff} is the diffusion coefficient.

Specific Comments R1-12: *Why do the authors not show the fits of TGA experiments? How did the authors take into account coke formation for fitting the obtained TGA profiles? According to Extended Data Figure 4a,c,e,g, the overall catalyst weight increases with time on propane stream?*

Response: We thank the reviewer for this reminder and comment. We have supplemented the fitted curves in the TGA experiments. The weight loss in TG profiles should be caused by the release of

lattice oxygen. However, coke would form that caused the weight increase when the lattice oxygen was fully released, which is dominated in the sole vanadia catalysts, especially at high temperature (625 °C). Comparatively, for ceria and ceria-vanadia redox catalysts, adequate and stable oxygen release suppressed the formation of coke formation from 575 °C to 600 °C. Therefore, to weaken the influence of coke formation and estimate the kinetics of oxygen release, we fitted the data by considering the stage of weight loss on ceria (30CeAl) and ceria-vanadia (6V/30CeAl) at the temperature range from 575 °C to 600 °C within 120 min, as shown in Extended Data Figure 4b-f.

Extended Data Figure 5 | Oxygen release kinetics determined by thermogravimetry relaxation experiments. Isothermal time dependence of the mass loss and relaxation curve in the form of fractional weight: (a, b) at 550 °C, (c, d) 575 °C, (e, f) 600 °C, and (g, h) 625 °C in a mixture of 20% C₃H₈/He (10 mL/min) for CeO₂ (30CeAl), VO_x (6V/Al) and VO_x-CeO₂ (6V/30CeAl). **The dash lines represent the fitted thermogravimetry relaxation curves.**

Specific Comments R1-13: Pages 33, 34. Something is wrong with Extended Data Table 1?

Response: We thank the reviewer for the comment. We have carefully revised the Extended Data Table 1 and removed the fitted k_{chem} and D_{diff} for sole well-dispersed vanadia catalysts (6V/Al). In the approximation, reduction–oxidation experiments are performed with powders, which are characterized by an average diameter of the powder particles, so that it is useful to model the reduction–oxidation kinetics of a sphere of radius R investigated by Crank (J. Crank, The Mathematics of Diffusion, Oxford University Press, 1975.). However, due to the limited oxygen capacity (0.16 wt% at 600 °C) and well monolayer dispersed VO_x on the surface of Al₂O₃ in sole vanadia catalysts (6V/Al), the lattice oxygen could be rapidly released, which did not follow the oxygen diffusion in the sphere.

Additionally, when oxygen was fully released, coke was formed and dominated in the sole vanadia catalyst (6V/Al) to significantly influence the estimation of the oxygen release kinetics. Therefore, for the comparisons of k_{chem} and D_{diff} determined by the TGA fittings, we just focused on the samples of ceria (30CeAl) and ceria-vanadia (6V/30CeAl) from 550 to 600 °C without significant coke formation to investigate the contribution of lattice oxygen bulk diffusion and surface reaction from ceria-vanadia interaction. The fitted k_{chem} and D_{diff} for ceria (30CeAl) and ceria-vanadia (6V/30CeAl) over a temperature range from 550 to 600 °C were shown in Extended Data Figure 6.

Extended Data Table 1. The surface reaction coefficient (k_{chem}) and bulk diffusion coefficient (D_{diff}) of oxygen in CeO₂ (30CeAl) and VO_x-CeO₂ (6V/30CeAl) at different temperatures were determined by analyzing the thermogravimetric relaxation kinetics.

T (°C)	VO _x	k_{chem} (cm/s)		VO _x	D_{diff} (cm ² /s)	
		CeO ₂	VO _x -CeO ₂		CeO ₂	VO _x -CeO ₂
550	-	9.23E-07	9.07E-06	-	2.51E-06	2.07E-05
575	-	2.23E-06	1.12E-05	-	3.66E-06	4.21E-05
600	-	4.00E-06	1.93E-05	-	1.14E-05	6.21E-05

Extended Data Figure 6 | The Arrhenius plots of k_{chem} and D_{diff} from 550 to 600 °C. Arrhenius plots of the (a) k_{chem} for CeO₂ (30CeAl) and VO_x-CeO₂ (6V/30CeAl) and (b) D_{diff} for CeO₂ (30CeAl) and VO_x-CeO₂ (6V/30CeAl) over a temperature range from 550 to 600 °C.

Specific Comments R1-14: Captions of some figures are confusing and do not contain catalyst composition. Please, check the manuscript?

Response: We thank the reviewer for this positive evaluation. We have carefully revised the captions and supplemented the catalyst information throughout the manuscript, such as formula, composition, and so on.

In page 4

The core-shell redox catalysts were prepared using a modified two-step incipient wetness impregnation method. The ceria-vanadia samples were named as $xV/yCeAl$, where $x(y)$ is the percent weight of V(Ce). The vanadia and ceria catalysts were obtained by VO_x and CeO_2 supported on $\gamma-Al_2O_3$ with the same V and Ce percent weights with VO_x-CeO_2 , respectively...

Fig. 1 | Identification of vanadia layers that coat ceria nanodomains. (a) Diagram of core-shell redox catalysts in propane dehydrogenation by chemical looping engineering: dehydrogenation and oxidation in fuel reactor (reducer) and air reactor (oxidizer), respectively. (b-d) HAADF-STEM images and (e-h) EELS mappings of core-shell ceria-vanadia redox catalysts (6V/30CeAl): (f): V; (g): Ce; (h): V + Ce. (i) Line-scanning EELS. (j) EELS of the domains ((1), (2), (3)) in (b).

Fig. 2 | Chemical looping oxidative dehydrogenation performance. (a) in situ XRD patterns of ceria-vanadia redox catalysts (6V/30CeAl). (b) Comparison of ceria (30CeAl), vanadia (6V/Al), and ceria-vanadia redox catalysts (6V/30CeAl). Conditions: 600 °C, GHSV = 2,500 h⁻¹, C₃H₈/N₂=0.25. (c) Comparing ceria-vanadia redox catalysts (6V/30CeAl) with established oxide-based and Pt-containing catalysts (see Extended Data Table 2). Motifs of triangle, rhombus, and sphere represent ODH, PDH, and CL-ODH, respectively. (d) Cyclic performance over ceria-vanadia redox catalysts (6V/30CeAl). Dehydrogenation step: 600 °C, GHSV = 2,500 h⁻¹, C₃H₈/N₂=0.25 for 30 minutes; Inert purge: 600 °C, N₂=40 mL/min for 5 minutes; Oxidation step: 600 °C, 20 vol.% O₂/N₂=20 mL/min for 15 minutes. (e) Comparison of energy consumption and CO₂ emission of traditional Oleflex technology and CL-ODH (see Methods and simulation settings in Supplementary Tables 5-7).

Fig. 3 | Experimental evidence of oxygen diffusion and surface reaction. (a) In situ XANES spectra of Ce L₃-edge (CeO₂ standards as the references) over ceria-vanadia (6V/30CeAl) (top) and pure ceria (30CeAl) (bottom) and (b) V K-edge (V foil, V₂O₅, VO₂, and V₂O₃ standards as the references) over ceria-vanadia (6V/30CeAl) (top) and pure vanadia (6V/Al) (bottom) at 600 °C under the flow of 20% C₃H₈/N₂ (20 mL/min). (c) In situ Raman spectra of ceria-vanadia (6V/30CeAl) (top) and vanadia (6V/Al) (bottom) at 600 °C under the flow of 20% C₃H₈/N₂ (20 mL/min). In situ DRIFTS spectra of temperature-programmed and isothermal propane dehydrogenation over ceria-vanadia (6V/30CeAl) (d) and vanadia (6V/Al) (e, f).

Specific Comments R1-15: How can NH₃-TPD tests be used to determine the concentration of Lewis acidic sites exclusively?

Response: We thank the reviewer for this valuable comment. We have revised the relevant statement on the measurement and analysis of acidic sites in the main text. Although TPD of

ammonia is a useful method for characterizing the acidic properties and acidic concentration, it is difficult to distinguish the Lewis acidic sites and Brønsted acid sites. Based on the different temperature ranges of the weak (120–200 °C), medium (200–350 °C), and strong acidic sites (350–450 °C) (Appl. Catal. A 1998, 168, 373-384; ACS Catal. 2015, 5, 438-447.), we think that the presence of surface VO_x sites increased the medium acidic sites.

In page 10

... Gradually decreased ratios of H₂/H₂O in pure ceria implied surface H abstraction were sluggish due to the less active C–H dissociation center of CeO₂. We then designed a temperature-programmed surface reaction (TPSR) to explore the surface reaction route. Initial dehydrogenation temperatures of C₃H₈ (m/z=29) on pure vanadia and ceria-vanadia redox catalysts were at 338 and 340 °C, respectively, lower than that of pure ceria (412 °C) (Extended Data Fig. 3), indicating the presence of low-temperature C–H dissociation V–O centers ascribed from active vanadia catalysts, corresponding to increased medium acidic sites as shown in NH₃-TPD profiles (Supplementary Fig. 11).

Specific Comments R1-16: According to a recent review from the same research group (DOI: 10.1039/d0cs00814a), there are several Pt-containing non-oxidative dehydrogenation catalysts which showed very high propene selectivity at high (above 50%) degrees of propane conversion but are not mentioned in Fig.2c. Some relevant reference should be reported.

Response: We thank the reviewer for this comment. In addition to the reported oxide-based catalysts, Pt-containing non-oxidative dehydrogenation catalysts have been included in the updated results in Fig. 2c and Extended Data Table 2.

Figure. R1-4 Comparing ceria-vanadia redox catalysts (6V/30CeAl) with established oxide-based and Pt-containing catalysts (see Extended Data Table 2). Motifs of triangle, rhombus, and sphere represent ODH, PDH, and CL-ODH, respectively.

Extended Data Table 2. Comparison of VO_x-CeO₂ catalysts with state-of-the-art ODH, PDH and CL-ODH catalysts.

Number	Sample	Temp (°C)	Model	Ratio (C ₃ H ₈ :O ₂)	P _{C₃H₈} (kPa)	C ₃ H ₈ Conv. (%) ^b	C ₃ H ₆ sel. (%) ^b	Ref.
1	PtPd/Al ₂ O ₃ /SiO ₂	450	ODH	2:1	10	32.4	17.0	47
2	Pd/MgVO	450	ODH	3.5:1	14	25.0	32.0	48
3	BS-1	570	ODH	1:1	16.6	43.7	53.3	49
4	BNNT	490	ODH	2:1	30	16.0	~75.0	5
5	Cr ₂ O ₃ /SBA-15	450	ODH	1:1	2.5	29.7	16.0	50
6	MoO _x /Al ₂ O ₃	500	ODH	1:1	4	12.9	27.0	51
7	MoO _x /Al ₂ O ₃	550	ODH	1:1	4	32.4	27.3	51
8	VO _x /SBA-15	600	ODH	2:1	10	19.6	41.0	52
9	VO _x /Al ₂ O ₃	500	ODH	2:1	10	~18	~40	53
10	VO _x /TiO _x /SiO ₂	500	ODH	2:1	~10	9.5	50.0	54
11	VO _x /ZrO ₂	500	ODH	1:1	~10	33.1	17.1	55
12	VO _x /graphene	500	ODH	2:1	16.7	5.0	92.1	56
13	(Pt/Al ₂ O ₃)@35cIn ₂ O ₃	450	ODH	2:1	10	40	75.6	6
14	CrO _x /SBA-1	550	PDH	-	6.7	37	85	57
15	Cr-Al-800	600	PDH	-	101	33.2	90.4	22
16	Cr ₁₀ Zr ₉₀ O _x	550	PDH	-	40.4	30	82	4
17	VO _x /γ-Al ₂ O ₃	600	PDH	-	~39	25	72	28
18	VO _x /SiBeta	600	PDH	-	5	38	88	58
19	VO _x /MCM-41	550	PDH	-	30.4	22	92	59
20	VO _x /(SiO ₂ +Al ₂ O ₃)	550	PDH	-	36	31	80	60
21	VO _x /(SiO ₂ +Al ₂ O ₃)	550	PDH	-	48	35	80	29
22	VO _x -K/meso-Al ₂ O ₃	610	PDH	-	28	33	83	61
23	VO _x /γ-Al ₂ O ₃	600	PDH	-	28	32	94	27
24	PtSn/MgAl ₂ O ₄	580	PDH	-	10	46.4	99.5	62
25	PtSn/MgSBA-15	580	PDH	-	70	43	97.8	63
26	PtSn/Al ₂ O ₃ (A750)	590	PDH	-	16	49	97	64
27	Pt/Sn-Beta	570	PDH	-	10	50.5	92.5	65
28	Pt-Sn-ZSM	550	CL-ODH	-	~101	~23	98	66
29	V-Mo-O oxide	500	CL-ODH	-	28.4	~38	~80	11
30	LaNiO _x	550	CL-ODH	-	10.1	11	40	67
31	VO _x /CaO-γAl ₂ O ₃	640	CL-ODH	-	-	25	94	68
32	CeO ₂ /Al ₂ O ₃	600	CL-ODH	-	28.4	10.5 ^a	81 ^a	This work
33	VO _x /Al ₂ O ₃	600	CL-ODH	-	28.4	36 ^a	78 ^a	This work
34	VO _x /CeO ₂ -Al ₂ O ₃ redox catalyst	600	CL-ODH	-	28.4	49 ^a	89 ^a	This work

Specific Comments R1-17: *The conclusion paragraph is too general and does not seem to be really representative for this study. What does this phrase “diffusion and reaction of lattice oxygen species was demonstrated and kinetically coordinated” mean? Why and how kinetically coordinated? The authors have written “further improvements are expected for their exact atomic and electronic structure.” Which improvements are expected? How can they be achieved? Similar sentences are also written in the abstract. I am also wondering why the authors have written in the abstract “Propane dehydrogenation (PDH) has the potential to produce industrially important platform chemicals.” Why does this reaction have the potential? Non-oxidative propane dehydrogenation is the basis of several large-scale processes.*

Response: We thank the reviewer for this valuable evaluation. We have carefully revised the abstraction, conclusion and improved the article accordingly.

ABSTRACT: *Propane dehydrogenation (PDH) is an industrial technology for direct propylene production which has received extensive attention in recent years. Nevertheless, existing non-oxidative dehydrogenation technologies still suffer from the thermodynamic equilibrium limitations and severe coking. Here, we developed the intensified propane dehydrogenation to propylene by the chemical looping engineering on nanoscale core-shell redox catalysts. The core-shell redox catalyst combined dehydrogenation catalyst and solid oxygen carrier at one particle, preferably composed of two to three atomic layer-type vanadia coating ceria nanodomains. The highest 93.5% propylene selectivity was obtained, sustaining 43.6% propylene yield under 300 long-term dehydrogenation-oxidation cycles, which outperforms an analog of industrially relevant $K-CrO_x/Al_2O_3$ catalysts and exhibits 45% energy savings in the scale-up of chemical looping scheme. Combining in situ spectroscopies, kinetics, and theoretical calculation, an intrinsically dynamic lattice oxygen “donator-acceptor” process was proposed that O^{2-} generated from the ceria oxygen carrier was boosted to diffuse and transfer to vanadia dehydrogenation sites via a concerted hopping pathway at the interface, stabilizing surface vanadia with moderate oxygen coverage at pseudo steady state for selective dehydrogenation without significant overoxidation or cracking.*

Conclusion: *In conclusion, the ceria-vanadia core-shell redox catalysts have been designed for propane dehydrogenation via the chemical looping engineering, which exhibited highest 93.5% propylene selectivity and 43.6% propylene yield during the long-term dehydrogenation-oxidation cycles. An intrinsically dynamic lattice oxygen “donator-acceptor” process in core-shell redox catalyst was proposed by the combination of in situ XAS, Raman, transient pulses and oxygen release kinetic analysis, that the lattice oxygen generated from ceria were boosted to diffuse (close to one order of magnitude higher D_{diff} at 550–600 °C than that of ceria oxygen carrier), and transfer to surface vanadia dehydrogenation sites via a concerted hopping pathway at the interface, stabilizing surface vanadia for selective dehydrogenation without significant overoxidation or cracking. Atomic-level details of oxygen diffusion and surface reaction over ceria-vanadia redox catalysts were further revealed by DFT calculations that derived from the structure of $ML-VO_2$ on CeO_2 , O^{2-} diffusion from $O(Ce)$ and subsequent transfer from interface $V-O-Ce$ to $V=O$ mediated by bridge $V-O-V$ exhibited the lowest barrier lower than that of direct and isolated hopping pathway. This study opens the possibilities for the design and application of efficient core-shell redox catalysts in selective oxidations and chemical looping systems.*

Reviewer #2:

General Comments R2: *This manuscript written by Jinlong Gong et al. proposes ceria-vanadia core-shell catalysts as promising redox catalysts for propane dehydrogenation. The authors state that charged oxygen ions (O^{2-}) generated from ceria oxygen carrier promote catalytic performance of ceria-vanadia catalysts for this reaction. Reliable characterization and adequate analyses are given in the manuscript. After reviewing this paper, I recommend that this paper is possible to be published in Nature Communications subjected to minor revisions. Some parts needed to be clarified.*

Response: We thank the reviewer for the kind comments. We have revised this manuscript carefully and improved the quality according to the comments.

Specific Comments R2-1: *In abstract, the authors state that “Charged oxygen ions (O^{2-}) generated from ceria oxygen carrier was boosted to diffuse and transfer to vanadia dehydrogenation sites via a concerted hopping pathway at interface, stabilizing surface vanadia with moderate oxygen coverage at pseudo steady state for selective oxidative dehydrogenation without overoxidation or cracking.”. Using “without overoxidation or cracking” is too strong, this means no overoxidation or cracking should be observed. However, coking was observed in a ceria-vanadia catalyst as presented in Fig.3c.*

Response: We thank the reviewer for this precious comments. Indeed, with the release of lattice oxygen, the transformation from oxidative dehydrogenation to non-oxidative dehydrogenation and cracking occurred. Based on the reviewer’s suggestion, we have revised the statement “without overoxidation or cracking” to “without significant overoxidation or cracking”.

ABSTRACT: *Propane dehydrogenation (PDH) is an industrial technology for direct propylene production which has received extensive attention in recent years. Nevertheless, existing non-oxidative dehydrogenation technologies still suffer from the thermodynamic equilibrium limitations and severe coking. Here, we developed the intensified propane dehydrogenation to propylene by the chemical looping engineering on nanoscale core-shell redox catalysts. The core-shell redox catalyst combined dehydrogenation catalyst and solid oxygen carrier at one particle, preferably composed of two to three atomic layer-type vanadia coating ceria nanodomains. The highest 93.5% propylene selectivity was obtained, sustaining 43.6% propylene yield under 300 long-term dehydrogenation-oxidation cycles, which outperforms an analog of industrially relevant $K-CrO_x/Al_2O_3$ catalysts and exhibits 45% energy savings in the scale-up of chemical looping scheme. Combining in situ spectroscopies, kinetics, and theoretical calculation, an intrinsically dynamic lattice oxygen “donator-acceptor” process was proposed that O^{2-} generated from the ceria oxygen carrier was boosted to diffuse and transfer to vanadia dehydrogenation sites via a concerted hopping pathway at the interface, stabilizing surface vanadia with moderate oxygen coverage at pseudo steady state for selective dehydrogenation without significant overoxidation or cracking.*

Specific Comments R2-2: On page 3, “Electron energy loss spectra (EELS) mappings of V L_{2,3} and Ce M_{4,5} edges affirmed vanadia sites anchored on widespread ceria nanodomains (Fig. 1h-k, Supplementary Fig. 1e-h).”. There is no Fig. 1k in the main text.

Response: We are grateful to the reviewer for the reminders. We have revised the captions in the figures and main text.

In page 5

At vanadia surface density of 4.3 V/nm² (Supplementary Table S1)¹⁷⁻¹⁹ for ceria-vanadia redox catalysts (6V/30CeAl), atom-resolved high-angle annular dark-field scanning transmission electron microscope (HAADF-STEM) images identified vanadia mainly existed as monolayers and bilayers along ceria surface (Fig. 1b-d). **Electron energy loss spectra (EELS) mappings of V L_{2,3} and Ce M_{4,5} edges affirmed vanadia sites anchored on widespread ceria nanodomains (Fig. 1e-h).**

Specific Comments R2-3: On page 10, the authors explain that “When surface oxygen was fully removed and ML-VO₂ formed, oxygen diffusion from CeO₂ to VO_x turned to be endothermic with reaction energy of 0.62 eV. However, it became exothermic with further oxygen release in VO_x between ML-VO₂ and ML-V₂O₃, verifying continuous oxygen transfer from O(Ce) to O(V) sites (Supplementary Fig. 13).” How Supplementary Fig. 13 could explain “exothermic” and verify “continuous oxygen transfer from O(Ce) to O(V) sites”?

Response: We thank the reviewer for the valuable comment. The “exothermic” means the reaction that transfers an oxygen atom from CeO₂ to vanadia and leaves a vacancy in CeO₂ is exothermic. We calculated the energy difference between initial state and final state of this reaction. The initial state was the formation of one oxygen vacancy in ML-VO_x and the final state was the oxygen vacancy fulfilled by the oxygen from CeO₂ that induced the formation of an oxygen vacancy in CeO₂. Bader charge analysis (Fig. S13 and Table S10) showed that, in both ML-V₂O₅ and ML-VO₂ models, the calculated valency electrons number of V atoms are same as the results in V₂O₅, which denotes the V atoms preserve their valency states and CeO₂ is reduced instead. The experimental results and in situ XAS spectra also showed that ceria in ceria-vanadia redox catalysts were reduced and acted as an “oxygen reservoir” that donated its lattice oxygen from bulk to stabilize surface vanadia. Hence, the thermodynamics of oxygen diffusion from O(Ce) to O(V) could be exothermic and the continuous oxygen transfer from ceria bulk would occur.

Specific Comments R2-4: There are few comments for the DFT part.

- Propylene seems to be strongly attached on the catalyst’s surface. How much are the desorption energies required for releasing propylene from the ceria-vanadia and V₂O₅ catalysts?
- The mechanistic studies of OPDH in V₂O₅ were reported. The authors should compare the results with literature. In addition, the authors investigated the stepwise mechanism. The authors should consider the concerted mechanism as well, especially in the ML-VO₂ model.
- The electronic charge analyses such as Bader charge or density of state (DOS) should be conducted to understand the valence states of atoms and electronic nature of studied catalysts.

- What is a criterion for the electronic convergence used in this work?

Response: We thank the reviewer for the valuable comments. We have supplemented the relevant calculations.

(1) We have calculated the adsorption energy of propylene to show the desorption tendency. The adsorption energies of propylene are determined to be -0.63 eV on ML-VO₂ and -0.37 eV on V₂O₅, respectively. The adsorption of propylene on ML-VO₂ is slightly stronger than V₂O₅, but the difference is smaller than the difference between activation barriers of side products and C₃H₆ formation. Therefore, ML-VO₂ possesses higher propylene selectivity than V₂O₅.

(2) We compared our calculated result on V₂O₅ with the results reported in *ACS Catal.* 2019, 9, 5816–5827. The energy profiles of propane dehydrogenation on V₂O₅ were shown in Figure 5(c) in the literature and the calculated activation barriers of first and second dehydrogenation steps were 1.18 eV and 1.57 eV. These values are very comparable with our results (1.12 eV and 1.60 eV).

Figure 5(c) of *ACS Catal.* 2019, 9, 5816–5827.

The reviewer suggests considering the concerted mechanism in ML-VO₂, we want to highlight the proposed concerted oxygen transfer mechanism is calculated in ML-VO₂ model indeed. Here we supplemented the concerted reaction on the surface, formation of H₂O(g) and H₂(g) in the models of ML-VO₂ and ML-V₂O₃ as shown in Figure S16. In the model of ML-VO₂, the main adsorption site of H* is V=O site. The activation barrier of formation H₂* intermediate is low while the formed H₂* has a stronger preference to form H₂O(g) rather than H₂(g). Such results denote ML-VO₂ is still in the dominant period of oxidative dehydrogenation. When the vanadia is further reduced, ML-V₂O₃ was existed. In Figure S16b, we could observe that all V=O sites are removed, and main adsorption site of H* becomes V-O-V. The formation activation barrier of H₂* is higher compared with the barrier in ML-VO₂, indicating the elimination of H* and regeneration of active site are more difficult. Further, H₂* tends to form H₂(g) in this period, denoting the model already enters the dominant period of non-oxidative dehydrogenation, according to our experimental

results that in the initial period with sufficient lattice oxygen, propane oxidative dehydrogenation was dominated, however when the lattice oxygen was fully reduced, non-oxidative dehydrogenation would be dominated.

Figure S16. Reaction energy profiles of H₂O formation and H₂ formation over (a) ML-VO₂ and (b) ML-V₂O₃.

(3) We have supplemented the Bader charge analysis to further prove our conclusion about CeO₂ was reduced and vanadia preserved its valency state before ML-V₂O₃ state. As shown in Figure S13, we marked the valency electrons difference of CeO₂ in ML-VO_x and pure CeO₂ to show the electrons accumulation on the CeO₂. The results indicated that in ML-V₂O₅, CeO₂ preserved its state and the reduction has not started in this model. In the following ML-VO₂ model, the CeO₂ started to receive electrons from VO_x (3.13 e⁻ in total) and each Ce atom showed more valence electrons compared with pure CeO₂, denoting CeO₂ was reduced. Finally, in ML-V₂O₃, the similar trend was observed, and more electrons were transferred from VO_x to CeO₂ (3.43 e⁻). Meantime, as shown in Table S10, in both ML-V₂O₅ and ML-VO₂ models, the calculated valency electrons number of V atoms are same as the results in V₂O₅, which denotes the V atoms preserve their valency states and Ce is reduced instead. The reduction of V atoms only happens in ML-V₂O₃ model.

Figure S13. Bader charge difference for CeO₂ in (a) ML-V₂O₅, (b) ML-VO₂ and (c) ML-V₂O₃. The Bader charge result for pure CeO₂ surface is taken as reference to show the accumulation of

electrons in ML-VO_x.

Table S10. Bader charge difference for V atoms in (a) ML-V₂O₅, (b) ML-VO₂ and (c) ML-V₂O₃. The Bader charge result for pure V₂O₅ surface is taken as reference to show the accumulation of electrons in ML-VO_x.

V atom number	ML-V ₂ O ₅ (e ⁻)	ML-VO ₂ (e ⁻)	ML-V ₂ O ₃ (e ⁻)
1	0.02	0.01	0.02
2	-0.03	0.01	0.26
3	-0.05	0.01	0.02
4	0.00	-0.01	0.27
5	0.02	0.01	0.02
6	-0.03	-0.03	0.26
7	-0.05	0.02	0.03
8	0.00	0.00	0.24
9	0.02	0.00	0.02
10	-0.03	-0.01	0.26
11	-0.05	0.01	0.03
12	0.00	0.00	0.27
13	0.02	0.01	0.02
14	-0.03	-0.02	0.25
15	-0.05	0.01	0.03
16	0.00	-0.02	0.26
Average	-0.02	0.00	0.14

(4) The criterion for the electronic convergence used in this work is 10⁻⁴ eV. We have revised the computational details in our manuscript.

In Page 12

Atomic-level details of oxygen diffusion and surface reaction over ceria-vanadia redox catalysts were investigated by density functional theory (DFT) calculations. With the elimination of O of VO_x (O(V)) in CeO₂-VO_x (Extended Data Fig. 7), the outermost O of CeO₂ (O(Ce)) started to coordinate with V to form V-O-Ce interface that exposes Ce³⁺ centers (Fig. 4a). Bader charge analysis further proves electrons accumulate on Ce with reduction of VO_x (Supplementary Fig. 13). Meantime, VO_x preserves its valency state until ML-V₂O₃ period (Supplementary Table 10). Existence of monolayer-V₂O₅ (ML-V₂O₅) on CeO₂ expectedly activated surface oxygen with lower oxygen vacancy (O_{vac}) formation energies than that of pure CeO₂. When surface oxygen was entirely removed, and ML-VO₂ formed,...

In Page 13

Valence states of V have been believed to manipulate propylene selectivity^{35,36}. The following calculations on the formation pathways of H₂O(g) and H₂(g) showed that ML-VO₂ has a strong preference to proceed oxidative dehydrogenation compared with ML-V₂O₃ (Supplementary Fig. 16) and exhibited lower barriers of first and second dehydrogenation of propane to propylene, but higher barriers for acetone formation, a significant intermediate of overoxidation, than crystalline V₂O₅ and ML-V₂O₃ (Fig. 4d, Extended Data Fig. 9), according to experimental results that ceria-vanadia redox catalysts showed higher initial propane conversion and propylene selectivity than 6V/Al and crystalline V₂O₅. These results firmly support that lattice oxygen transfer from ceria was to stabilize moderate V valence states and oxygen coverage for selective dehydrogenation without significant overoxidation or cracking and coking.

In Page 24

Computational details.... The relaxation of the electronic degrees of freedom will be stopped if the total energy change between two steps smaller than 10⁻⁴ eV.

Specific Comments R2-5: For the Fig.3c's caption, (top) and (bottom) should be specified in "(c) In situ Raman spectra of ceria-vanadia and vanadia at 600 °C". There are typos such as "transient pules" in Fig. 3's caption and the main text. In Fig. 4, each pathway should be labeled likes Extended Data Figure 8 or described in the caption?

Response: We are grateful to the reviewer for the reminders. We have revised the captions in Fig 3 and main text. Additionally, we have labeled each reaction pathway in Fig. 4.

In Page 10

We then designed a temperature-programmed surface reaction (TPSR) and transient pulses to explore the surface reaction route....

...Derived from Arrhenius plots of D_{diff} and k_{chem} at 550-625 °C, ..., which accounts for kinetic modulation between oxygen diffusion and surface reaction, as shown in transient pulse experiments.

Fig. 3 | Experimental evidence of oxygen diffusion and surface reaction. (a) In situ XANES spectra of Ce L₃-edge (CeO₂ standards as the references) over ceria-vanadia (top) and pure ceria (bottom) and (b) V K-edge (V foil, V₂O₅, VO₂, and V₂O₃ standards as the references) over ceria-vanadia (top) and pure vanadia (bottom) at 600 °C under the flow of 20% C₃H₈/N₂ (20 mL/min). (c) In situ Raman spectra of ceria-vanadia (top) and vanadia (bottom) at 600 °C under the flow of 20% C₃H₈/N₂ (20 mL/min).

In page 29

Extended Data Figure 4 | Surface reaction routes determined by the temperature-programmed surface reaction and transient pulse experiments. C₃H₈-temperature programmed surface reaction (TPSR), C₃H₈ pulse spectra at 600 °C and the calculated F_{H_2/H_2O} , which is the intensity ratio of H₂/H₂O obtained from the pulse profiles, (a-c) vanadia, (d-f) ceria-vanadia and (g-i) ceria catalysts. C₃H₈, C₃H₆, CO₂, CH₄, H₂, and H₂O, m/e equals 29, 41, 44, 16, 2 and 18, respectively.

Fig. 4 DFT calculations on oxygen diffusion and surface reaction. (a) Models of CeO₂ and ML-VO₂ (inset [V₄O₈] units). (b) Optimal concerted hopping pathway in ML-VO₂ contrast to (c) direct hopping pathway. (d) The calculated energy profiles on ML-VO₂ (blue) and crystalline V₂O₅ (yellow). Reaction steps include (i) dehydrogenation from propane to absorbed propyl, (ii) dehydrogenation from absorbed propyl to absorbed propene, and (iii) dehydrogenation from absorbed propyl to acetone. V: dark blue; Ce: bright blue; O: red; C: black; H: white.

Reviewer #3:

General Comments R3: *Chen and co-workers have carried out a comprehensive investigation of vanadia on ceria nanoparticles for the selective oxidative dehydrogenation of propane. This report is a multi-scale investigation as the authors address atomic and molecular aspects of the catalyst performance—such as oxygen transport from ceria to vanadia—to process-wide analysis and comparison to existing propane dehydrogenation processes. However, there have been multiple investigations of vanadia/ceria catalysts (by well-known groups such as Freund in 2010 (10.1021/ja910574h) to more recent reports that investigate the structure and function of vanadia/ceria in-situ and at the molecular level (see this report by X. Wu et al. (10.1021/jacs.5b07939), by Hess et al (10.1021/jacs.2c06303), Iglesias-Juez (10.1021/acs.jpcc.7b09832), Taylor et al (10.1007/s11244-009-9307-0), and Sauer (10.1021/jp108185y)).*

Response: We thank the reviewer for this positive evaluation. We have carefully addressed the issues raised by the reviewer and revised the manuscript carefully."As the references listed by the reviewer, vanadia/ceria catalysts have attracted increasing attention in the oxidative dehydrogenation of propane with O₂ co-feeding, indicating the importance and preferential application of vanadia/ceria catalysts in this process.

Different from the electronic effects and redox properties of reported vanadia/ceria catalysts following the Mars-van-Krevelen (MvK) mechanism with O₂-cofeeding, in this work we proposed the chemical looping process for the anaerobic oxidative propane dehydrogenation without O₂-cofeeding and revealed an intrinsically dynamic lattice oxygen “donator-acceptor” process via a concerted hopping pathway at the vanadia-ceria interface by constructing the core-shell ceria-vanadia redox catalysts, which was not reported in previous research. We believe that these mechanistic understandings will provide new insights into the design and application of redox catalysts for propane dehydrogenation via the chemical looping engineering.

To address the novelty of this work, the research status and remaining open questions of vanadia/ceria catalysts have been supplemented in the Introduction part.

In page 3

Chemical looping engineering offers exciting new opportunities for the challenges through physical or temporal separation of dehydrogenation and oxidation by solid oxygen carrier mediums^{9,10}. Unlike traditional catalysts, the oxygen carriers react with alkanes and undergo reversible changes by donating and replenishing oxygen to close the loop in the reducer and oxidizer reactors. Most oxygen carriers involve the metal centers or oxide composites to modulate lattice oxygen reactivity, using bulk doping¹¹, surface modification¹², or confinement in supports¹⁴. Recently, vanadia/ceria catalysts have attracted increased attention in the oxidative dehydrogenation of propane with O₂ co-feeding. The electronic effects and redox properties were investigated at the molecular level¹³⁻¹⁶. Nevertheless, direct experimental and theoretical insights into the lattice oxygen diffusion and the surface dynamics have not been reported yet for the anaerobic oxidative dehydrogenation via the chemical looping engineering.

Specific Comments R3-1: *Chen et al have found how to prepare composite vanadia on ceria particles that seem to have the right dimensions and composition to work effectively in the context of a propane dehydrogenation cyclic process, which is a significant technical achievement, but not really a scientific one. In my opinion, the latest implementation of the CATOFIN propane dehydrogenation process is, functionally, similar to the process described in Figure 1. This latest process uses a dehydrogenation catalyst (chromia on alumina) AND a so-called Heat Generating Material (which I don't know what it is, but functions like the ceria in this report, by consuming hydrogen and producing heat, (<https://www.clariant.com/en/Business-Units/Catalysts/Petrochemical-and-Refining-Catalysts/CATOFIN-Technology>)). Also, as described in Figure 1, the proposed process is not formally a 'looping' process because not the catalyst or the reactants 'loop' around the process. This is a typical case of process intensification run in a cyclic process?*

Response: We thank the reviewer for the important and positive comments for our technical achievement. We believe that the application of composite vanadia on ceria particles was not only a significant technical achievement, but also can provide further insights into the mechanistic relevance of oxygen diffusion and surface reaction in the core-shell redox catalysts for chemical looping propane dehydrogenation.

(1) In the view of scientific issue, combining in situ spectroscopies, kinetics, and theoretical calculation, we demonstrated and revealed that an intrinsically dynamic lattice oxygen “donator-acceptor” process via a concerted hopping pathway at the interface accounts for kinetic coordination between bulk diffusion and surface reaction in the core-shell redox catalyst, which was not reported in previous research. This “donator-acceptor” mechanism will provide new insights into the design and application of redox catalysts for propane dehydrogenation via chemical looping engineering.

(2) In terms of latest implementation of the CATOFIN propane dehydrogenation process using a dehydrogenation catalyst (chromia on alumina) and a so-called Heat Generating Material (HGM). We consider that two main differences of our core-shell redox catalysts from the Catofin process. Firstly, HGM in the Catofin process is a catalytically inert material that can not participate the dehydrogenation of propane while can store and generate the heat ($\text{H}_2 + \text{MO}_x = \text{MO}_{x-1} + \text{H}_2\text{O}$, $\Delta H < 0$) to make the temperature distribution in the catalysts bed more uniform (US7622623B2; US7973207B2). However, the ceria-vanadia redox catalysts investigated in this work are active for the oxidative dehydrogenation of propane that lattice oxygen from the ceria-vanadia redox catalysts involves the activation of C-H bonds of propane to produce propylene ($\text{C}_3\text{H}_8 + \text{MO}_x = \text{C}_3\text{H}_6 + \text{MO}_{x-1} + \text{H}_2\text{O}$). Secondly, HGM is physically mixed and loaded into the catalyst bed with the catalyst instead of chemically atomic contacting via the oxide interface in the ceria-vanadia core-shell redox catalysts in this work, which can provide the path for lattice oxygen diffusion and transfer at nanoscale. We have showed the ceria-vanadia core-shell redox catalysts (6V30Ce/Al) exhibited higher propylene selectivity and lower deactivation rate compared with the physical mixture of dehydrogenation catalyst VO_x (6V/Al) with solid oxygen carrier CeO_2 , denoted as 6V/Al+30Ce/Al (mass ratio 1:1) (Figure. R3-1), which performed well initially, but

performance decayed rapidly with time on stream, indicating the core-shell nanostructure allowing greater exposure of the VO_x sites that were better oxidized at the $\text{VO}_x\text{-CeO}_2$ oxide interface by the lattice oxygen from CeO_2 oxygen carriers. In situ characterizations and Bader charge analysis showed that V atoms preserve their oxidation valence states and CeO_2 is reduced instead. The designs in this work demonstrate that tight coupling between dehydrogenation catalytic sites and oxygen carriers could contribute to high stability that required the extensive and intimate connectivity afforded by the particular geometry of the overcoated nanostructure. The relevant results have been showed in Extended Data Figure 1c.

Figure. R3-1 Comparison of propane conversion and propylene selectivity on ceria-vanadia core-shell redox catalysts (6V30Ce/Al) and physical mixture of dehydrogenation catalyst VO_x (6V/Al) with solid oxygen carrier CeO_2 denoted as 6V/Al+30Ce/Al (mass ratio 1:1). Conditions: 600 °C, 1.4 atmospheric pressure, $\text{GHSV}=2500 \text{ h}^{-1}$, 0.5 g of sample, $\text{C}_3\text{H}_8/\text{N}_2 = 0.25$.

(3) The term “chemical looping” was first minted by Richter and Knoche in 1983 in the context of reducing exergy loss in fossil fuel combustion to intensify the process (Richter, H. J., & Knoche, K. F. (1983). Reversibility of combustion processes.). A typical chemical looping process involving oxygen-carrying agents is composed of two or more reduction and oxidation steps that form a redox loop. The concept of chemical looping has been extended well beyond combustion, such as chemical looping reforming (CLR), chemical looping partial oxidation (CLPO), chemical looping oxidative dehydrogenation (CL-ODH), and so on (*Nature Rev. Chem.* 2018, 2, 349–364; *Energy Environ. Sci.*, 2020, **13**, 772-804). In this work, ceria-vanadia redox catalysts serve as the oxygen carriers by donating lattice oxygen from oxidation states (V^{5+} , V^{4+} , Ce^{4+}) to propane in the dehydrogenation (reduction) step in the reducer, affording a reduced valence state (V^{3+} , Ce^{3+}) that can be reoxidized in the re-oxidation step by air in the oxidizer to close the loop as showed in Fig. 1a. To clarify the looping process, the function of ceria-vanadia redox catalysts (oxygen carriers) and application of chemical looping oxidative dehydrogenation of propane have been further introduced.

Extended Data Figure 1 | (c) Effect of Ce contents ranging from 0 to 30 wt% and physical mixture of vanadia (6V/Al) and ceria (30CeAl) catalysts on propane conversion and propylene selectivity over ceria-vanadia redox catalysts (6V/30CeAl). Reaction test conditions: 600 °C, 1.4 atmospheric pressure, GHSV=2500 h⁻¹, 0.5 g of sample, C₃H₈/N₂ = 0.25.

Chemical looping engineering offers exciting new opportunities for the challenges through physical or temporal separation of dehydrogenation and oxidation by solid oxygen carrier mediums^{9,10}. Unlike traditional catalysts, the oxygen carriers react with alkanes and undergo reversible changes by donating and replenishing oxygen to close the loop in the reducer and oxidizer reactors. Most oxygen carriers involve the metal centers or oxide composites to modulate lattice oxygen reactivity, using bulk doping¹¹, surface modification¹², or confinement in supports¹⁴. Recently, vanadia/ceria catalysts have attracted increased attention in the oxidative dehydrogenation of propane with O₂ co-feeding. The electronic effects and redox properties were investigated at the molecular level¹³⁻¹⁶. Nevertheless, direct experimental and theoretical insights into the lattice oxygen diffusion and the surface dynamics have not been reported yet for the anaerobic oxidative dehydrogenation via the chemical looping engineering.

To unravel the oxygen diffusion and reaction dynamics regarding the active sites, a nanoscale core-shell redox catalyst combining dehydrogenation catalyst and oxygen carrier at one particle was designed. The core-shell redox catalyst was preferably composed of 2-3 atomic layer-type vanadia coating ceria nanodomains to achieve the synergetic modulation of lattice oxygen bulk diffusion and surface reaction. In the dehydrogenation step (reducer), ceria-vanadia redox catalysts donate lattice oxygen for the dehydrogenation of propane to produce propylene, H₂O and H₂, affording a reduced valence state that can be reoxidized in the reoxidation step (oxidizer) by air to close the loop (Fig. 1a).

Specific Comments R3-2: *The paper is generally well written and logical in its presentation of information. The composite figures that are used throughout are not easy to follow and it would benefit from extending the description of the figures within the text (see Figure 3, for example). The text omits, in many instances, definite and indefinite articles and the readability of the paper would improve if this element were reviewed and corrected across the whole document.?*

Response: We thank the reviewer for the valuable comment. We have fully extended the descriptions of the figures within the text. Additionally, the articles and relevant sentences have been carefully and fully revised throughout the manuscript. Some representative revisions were showed here.

In page 8

In situ diffuse reflectance infrared Fourier transform spectroscopy (DRIFTS) upon propane exposure identified the co-existence of dehydrogenation and cracking of propane induced by this dynamic oxygen evolution. Peaks ascribed to asymmetric and symmetric CH₃ stretching modes (2,970 and 2,875 cm⁻¹) started at 100–150 °C (Fig. 3d)³⁰. The presence of a band centering at 1,645 cm⁻¹ (ν(CH₃CH=CH₂)) implied that the propyl complex was oxidatively dehydrogenated to propenyl by heterolytically subtracting H to neighboring V-O sites, leading to the occurrence of vanadium hydroxyl band (V-OH, 3,660 cm⁻¹)²⁸. However, a peak of ν(C=O) (1,680 cm⁻¹) attributed to acetone, the intermediate of overoxidation of propane to CO_x, was dominated on the pure VO_x catalysts when the temperature was higher than 150 °C, along with a significantly negative V=O band induced by the ready reduction of vanadia (V⁵⁺→V³⁺) (Fig. 3e, f and Supplementary Fig. 10)^{11,28}. This “over quick” oxygen removal would induce the transformation of oxidative to non-oxidative dehydrogenation and occurrence of propane cracking. At 250–600 °C, two peaks at 1545 and 1460 cm⁻¹ attributed to the unsaturated or aromatic species, the precursors of coke deposits^{27,28} that lead to fast deactivation were observed, which were also evidenced by the more dominated D₁ and G band in situ Raman spectra on pure vanadia catalysts during the dehydrogenation step.

In page 10

Evidenced by the in situ spectroscopies, the release of lattice oxygen would induce the existence and transformation of different reaction periods, including overoxidation, oxidative dehydrogenation, and non-oxidative dehydrogenation. Under differential reactor operation by controlling the propane conversion lower than 10%, C₃H₆ formation rate showed a linear relationship with C₃H₈ pressure while C₃H₈ conversion kept identical at different C₃H₈ pressures, indicating that the rate of propene formation in this reaction is typically directly related to propane partial pressure, i.e., a first-order reaction with respect to propane. To clarify the contribution of oxidative and non-oxidative dehydrogenation, the formation of H₂O and H₂ over ceria-vanadia redox catalysts in their dehydrogenation tests were investigated. As shown in Extended Fig. 3, the initial ratio of H₂O to H₂ at 5th min was determined to be 0.44, however, it would decrease to be about 0.05 after 60 min. Therefore, oxidative dehydrogenation was dominated in the initial period

and it was transformed to non-oxidative dehydrogenation with reaction time, accounting for the introduction of re-oxidation step to recover the lattice oxygen after 30-min dehydrogenation test during the continuous dehydrogenation-reoxidation cycles with the ratios of H₂O to H₂ of ~ 0.21.

To catch the transient distributions of products, C₃H₆, CO_x, H₂O and H₂, especially in the initial period, C₃H₈ transient pulse experiments were employed at 600 °C using on-line mass spectrometry (MS) (Extended Data Figure 4). CO_x was firstly observed due to the overoxidation on the active surface oxygen species (period I), then it would undergo selectively oxidative dehydrogenation to C₃H₆ and H₂O caused by lattice oxygen (period II). After the lattice oxygen were fully released, cracking or hydrogenolysis occurred which was more dominated for pure vanadia catalysts, leading to the formation of CH₄ (period III). Intensity ratios of H₂/H₂O defined as F_{H_2/H_2O} , a sign of H combustion, maintained at five over ceria-vanadia redox catalysts (Fig. 3g), implying that O²⁻ diffusion from ceria oxygen carrier leads to a pseudo-steady-state H combustion at surface vanadia sites^{6,31}. In contrast, pure vanadia showed stage-divided products, along with the tenfold increased intensity ratios of H₂/H₂O with pulses of C₃H₈, indicating the presence of O²⁻ gradient and transport limitation in the pure vanadia³¹. Gradually decreased ratios of H₂/H₂O in pure ceria implied surface H abstraction were sluggish due to the less active C–H dissociation center of CeO₂. We then designed a temperature-programmed surface reaction (TPSR) to explore the surface reaction route. Initial dehydrogenation temperatures of C₃H₈ ($m/z=29$) on pure vanadia and ceria-vanadia redox catalysts were at 338 and 340 °C, respectively, lower than that of pure ceria (412 °C) (Extended Data Fig. 3), indicating the presence of low-temperature C–H dissociation V–O centers ascribed from active vanadia catalysts, corresponding to increased medium acidic sites as shown in NH₃-TPD profiles (Supplementary Fig. 11).

Propane dehydrogenation (PDH) is an industrial technology for direct propylene production which has received extensive attention in recent years. Nevertheless, existing non-oxidative dehydrogenation technologies still suffer from the thermodynamic equilibrium limitations and severe coking. Here, we developed the oxidative dehydrogenation-like propane to propylene by chemical looping engineering on nanoscale core-shell redox catalysts. Xxx

Propane dehydrogenation (PDH) is an industrially important alternative to oil-based cracking processes^{1,2}. However, the commercial non-oxidative propane dehydrogenation containing CrO_x or Pt-based catalysts is endothermic and equilibrium-limited, necessitating much heat to achieve viable propylene yield^{3,4}. Although the oxidative dehydrogenation of propane (ODH) has the potential to improve conversion for favorable thermodynamics, propylene selectivity is hampered by overoxidation to CO₂^{5,6}. A similar challenge faces in selective oxidation reactions in the chemical industry^{7,8}.

Chemical looping oxidative dehydrogenation (CL-ODH) offers exciting new opportunities for the challenges through physical or temporal separation of dehydrogenation and oxidation by the solid oxygen carrier mediums^{9,10}. Unlike traditional catalysts, the oxygen carriers react with alkanes and undergo reversible changes by donating and replenishing oxygen to close the loop.

REVIEWERS' COMMENTS

Reviewer #1 (Remarks to the Author):

The authors have clarified a major part of my concerns. A few comments still need to be clarified.

i) Equations 1-2a and 1-2b are still mixed up. Equation 1-2a is used to calculate the selectivity to propene under consideration of coke formation but not equation 1-2b as written in the manuscript. I am also unsure if equation 1-2b is correct. Why does the numerator contain factor of "3"? What does n_i in the denominator mean?

ii) The answer to my original comment 3 in the Summary of revisions is not readable "We have clarified the dthe supplemented the initial conversion and formation rate of propylene at 5th min under differential reactor operation."

iii) If the initial ratio of H₂O to H₂ at 5th min was 0.44, is it really possible to conclude that "oxidative dehydrogenation was dominated in the initial period"?

Reviewer #2 (Remarks to the Author):

In the revised manuscript, the authors addressed all comments properly. After reviewing the revision of this paper, I recommend this paper to publish in Nature Communications.

Reviewer #1:

General comments R1: *The authors have clarified a major part of my concerns. A few comments still need to be clarified.*

Response: We thank the reviewer for this positive evaluation. We have carefully addressed the comments raised by the reviewer and revised the manuscript accordingly.

Specific Comments R1-1: *Equations 1-2a and 1-2b are still mixed up. Equation 1-2a is used to calculate the selectivity to propene under consideration of coke formation but not equation 1-2b as written in the manuscript. I am also unsure if equation 1-2b is correct. Why does the numerator contain factor of “3”? What does n_i in the denominator mean?*

Response: We apologized for the mixed Equations 1-2a and 1-2b. We have corrected them in the manuscript. Equation 1-2a is used to calculate the selectivity to propene under consideration of coke formation.

$$\text{Sel (\%)} = 100 \times [\text{F}_{\text{C}_3\text{H}_6}]_{\text{outlet}} / ([\text{F}_{\text{C}_3\text{H}_8}]_{\text{inlet}} - [\text{F}_{\text{C}_3\text{H}_8}]_{\text{outlet}}) \quad (1-2a)$$

$$\text{Sel (\%)} = 100 \times 3 \times [\text{F}_{\text{C}_3\text{H}_6}]_{\text{outlet}} / (\sum n_i \times [\text{F}_i]_{\text{outlet}}) \quad (1-2b)$$

For Equations 1-2b, C_3H_6 selectivity was calculated based on gas phase products. Due to the existence of side products i , such as CO_2 , CO , CH_4 , C_2H_6 , C_2H_4 , the C_3H_6 gas selectivity is normalized by the number of carbon atoms of component i . i stands for different hydrocarbon products in exhaust gases, n_i is the number of carbon atoms of side products i , (e.g., for CH_4 , $n_{\text{CH}_4}=1$, for C_2H_6 , $n_{\text{C}_2\text{H}_6}=2$). 3 is the number of carbon atoms of C_3H_6 .

In page 20

The instantaneous propane conversion and propylene selectivity based on all products (including coking formation) and gas phase products are defined as the instantaneous values at the different time on stream, according to equation (1-1) and equation (1-2) (2a: selectivity including coke formation and 2b: gas selectivity). The propylene yield was calculated based on propane conversion and propylene selectivity (including coke formation):

$$\text{Con (\%)} = 100 \times ([\text{F}_{\text{C}_3\text{H}_8}]_{\text{inlet}} - [\text{F}_{\text{C}_3\text{H}_8}]_{\text{outlet}}) / [\text{F}_{\text{C}_3\text{H}_8}]_{\text{inlet}}. \quad (1-1)$$

$$\text{Sel (\%)} = 100 \times [\text{F}_{\text{C}_3\text{H}_6}]_{\text{outlet}} / ([\text{F}_{\text{C}_3\text{H}_8}]_{\text{inlet}} - [\text{F}_{\text{C}_3\text{H}_8}]_{\text{outlet}}) \quad (1-2a)$$

$$\text{Sel (\%)} = 100 \times 3 \times [\text{F}_{\text{C}_3\text{H}_6}]_{\text{outlet}} / (\sum n_i \times [\text{F}_i]_{\text{outlet}}) \quad (1-2b)$$

$$\text{Yield (\%)} = \text{Con (\%)} \times \text{Sel (\%)} / 100 \quad (1-3)$$

Where i stands for different side products in exhaust gases, n_i is the number of carbon atoms of side products i , and F_i is the corresponding molar flow rate (mol/h). $[\text{F}_{\text{C}_3\text{H}_6}]_{\text{outlet}}$ is the flow of propylene out of the reactor (mol/h). $[\text{F}_{\text{C}_3\text{H}_8}]_{\text{outlet}}$ is the flow of propane out of the reactor (mol/h). $[\text{F}_{\text{C}_3\text{H}_8}]_{\text{inlet}}$ is the flow of propane in of reactor (mol/h). t is the time during the dehydrogenation stage (min). m is the weight of catalysts (g_{cat}). X_{initial} and X_{final} , respectively, represent the conversion measured at the initial and final period of an experiment. t

represents the reaction time (h). k_d is the deactivation rate constant (h^{-1}). A high k_d value means rapid deactivation, that is, low stability.

Specific Comments R1-2: *The answer to my original comment 3 in the Summary of revisions is not readable “We have clarified the dthe supplemented the initial conversion and formation rate of propylene at 5th min under differential reactor operation.”*

Response: We apologized for the typo errors in the comment 3 in the Summary of revisions. We meant that to investigate the dependence of propylene formation rates and propane conversion on propane pressure, we have supplemented the initial propene formation rate at 5th min under the differential operation by controlling the propane conversion lower than 10%.

Specific Comments R1-3: *If the initial ratio of H₂O to H₂ at 5th min was 0.44, is it really possible to conclude that “oxidative dehydrogenation was dominated in the initial period”?*

Response: We apologized for the misunderstandings. We have revised the sentence in the manuscript. Due to the detection methods used in GC, we can only detect the H₂O/H₂ after the initial 5 mins. Therefore, we meant that the ratios of H₂O to H₂ could be larger than 0.44 and oxidative dehydrogenation could be more dominated in the initial period less than 5 mins. We hope the quick GC detect method for the determination of H₂O/H₂ ratios can be developed in the near future.

In page 10

Evidenced by the in situ spectroscopies, the release of lattice oxygen would induce the existence and transformation of different reaction periods, including overoxidation, oxidative dehydrogenation, and non-oxidative dehydrogenation. Under differential reactor operation by controlling the propane conversion lower than 10%, C₃H₆ formation rate showed a linear relationship with C₃H₈ pressure while C₃H₈ conversion kept identical at different C₃H₈ pressures, indicating the rate of propene formation is typically directly related to propane partial pressure, i.e., a first-order reaction with respect to propane (Extended Fig. 3c). To clarify the contribution of oxidative and non-oxidative dehydrogenation, the formation of H₂O and H₂ over ceria-vanadia redox catalysts in their dehydrogenation tests were investigated. As shown in Extended Fig. 3, the initial ratio of H₂O to H₂ at 5th min was determined to be 0.44, however, it would decrease to be about 0.05 after 60 min. Therefore, oxidative dehydrogenation could be more dominated in the initial period (less than 5 mins) and it was transformed to non-oxidative dehydrogenation with time, accounting for the introduction of re-oxidation step to recover the lattice oxygen after 30-min dehydrogenation test during the continuous dehydrogenation-reoxidation cycles with the ratios of H₂O to H₂ of ~ 0.21.

Reviewer #2:

General comments R2: *In the revised manuscript, the authors addressed all comments properly. After reviewing the revision of this paper, I recommend this paper to publish in Nature Communications.*

Response: We thank the reviewer for this positive evaluation.